# FAST ENSEMBLING WITH DIFFUSION SCHRÖDINGER BRIDGE

**Hyunsu Kim**[*]**, Jongmin Yoon**[*]**, Juho Lee**
Kim Jaechul Graduate School of AI
KAIST
Daejeon, South Korea
{kim.hyunsu,jm.yoon,juholee}@kaist.ac.kr

## ABSTRACT

Deep Ensemble (DE) approach is a straightforward technique used to enhance the performance of deep neural networks by training them from different initial points, converging towards various local optima. However, a limitation of this methodology lies in its high computational overhead for inference, arising from the necessity to store numerous learned parameters and execute individual forward passes for each parameter during the inference stage. We propose a novel approach called Diffusion Bridge Network (DBN) to address this challenge. Based on the theory of the Schrödinger bridge, this method directly learns to simulate an Stochastic Differential Equation (SDE) that connects the output distribution of a single ensemble member to the output distribution of the ensembled model, allowing us to obtain ensemble prediction without having to invoke forward pass through all the ensemble models. By substituting the heavy ensembles with this lightweight neural network constructing DBN, we achieved inference with reduced computational cost while maintaining accuracy and uncertainty scores on benchmark datasets such as CIFAR-10, CIFAR-100, and TinyImageNet. Our implementation is available at https://github.com/kim-hyunsu/dbn.

## 1 INTRODUCTION

Deep Ensemble (DE) (Lakshminarayanan et al., 2017) is one of the straightforward yet powerful techniques for improving the performance of deep neural networks. This method involves averaging the outputs of multiple models trained independently with different random initializations and data scan orders. DE is an instance of bagging (Breiman, 1996) where the models are trained with fixed datasets, and can be interpreted as an approximatation of Bayesian Model Averaging (BMA). DE can significantly improve the prediction accuracy of deep neural networks, and more importantly, the uncertainty quantification and out-of-distribution robustness for tasks across various domain.

However, the major drawback of DE lies in the fact that the computational overhead and memory usage during inference calculations scale linearly with the number of ensemble members, which may be problematic for resource-limited environments or when a model is prohibitly large. Various approaches have been proposed to mitigate this issue, ideally reducing the inference cost of DE down to a single forward pass while minimizing the degradation in the predictive performance. A popular method in this direction is ensemble distillation, which is based on knowledge distillation (Hinton et al., 2015) over the ensemble outputs. In ensemble distillation, the average output of multiple ensemble members is set as the output of the teacher network, and a single model is set as a student whose output is learned to minimize the discrepancy from the teacher outputs. Additionally, techniques involving the use of multi-head models (Tran et al., 2021), shared weights (Wen et al., 2019), learning ensemble distributions (Malinin et al., 2020), employing diversity-promoting augmentation (Nam et al., 2021), or training generators simulating the ensemble predictions (Penso et al., 2022) have been presented. However, these methods either still demonstrate performance inferior to DE or require an inference cost comparable to DE in order to achieve equivalent performance.

Recently, Yun et al. (2023) proposed an orthogonal approach to reduce inference costs of DE. They achieve this by connecting ensemble members through low-loss subspaces, employing tech-

niques such as Bezier curves between each pair of ensemble members based on methods outlined by Garipov et al. (2018). For each ensemble member, a low-loss subspace is chosen, originating from that member, and a parameter within the subspace is sampled (usually from the center of the subspace). To streamline the inference process, they introduce Bridge Network (BN), a lightweight neural network that takes an intermediate feature from the ensemble member and directly predicts the output originally computed from the model on the low-loss subspace. The final output is approximated through an average of the ensemble member's output and the output predicted by BN. Crucially, BN is designed to have a negligible number of parameters and inference cost compared to the full ensemble model, resulting in significantly reduced inference costs. Experimental results using real-world image classification benchmarks and large-scale deep neural networks validate their approach, showcasing faster ensemble inference with sub-linear scaling of inference costs in relation to the number of ensemble members.

However, BN presented in Yun et al. (2023) comes with critical limitations. First, it does not directly predicts the output from the other ensemble members, but only for the models on the low-loss subspaces. This incurs an extra training cost of learning low-loss subspaces between all pairs of ensemble members. Second, a single BN can only be constructed between a pair of ensemble members. This means that as the number of ensemble members grow, the number of BNs to be constructed grows quadratically. Due to this structure, even though they achieved sub-linear growth in inference cost, there still is a large room for improvement.

In this paper, improving upon BN, we present a novel method for reducing inference costs of ensemble models. Given a set of ensemble members, we designate one of the ensemble members as a starting point. Then we train a mapping that transports an output from the starting point to the output computed from the ensemble model (averaged output). Compared to BN, our approach does not require learning low-loss subspaces, and do not require learning as many mappings as the number of pairs among ensemble members, at the cost of increased complexity to learn the mapping. Instead it learns this mapping via Diffusion Schrödinger Bridge (DSB) (Bortoli et al., 2021; Liu et al., 2023), a powerful model that can build a stochastic path between two probability distributions. Leveraging DSB, we create a sequence of predictions that progressively transition from the predictions of the starting point model to those of the ensemble model, capturing the inherent correlations among these predictions. Importantly, recognizing our primary objective of lowering inference costs, we design the score network of DSB to be lightweighted and incorporate diffusion step distillation to further minimize computational overhead. We refer to our approach as Diffusion Bridge Network (DBN), and their experimental results demonstrate significant improvements in the efficiency of ensemble inference cost reduction, surpassing the performance of BN and other ensemble distillation techniques.

## 2 BACKGROUNDS

### 2.1 DEEP ENSEMBLE AND BRIDGE NETWORK

In the ensemble methods as DE (Lakshminarayanan et al., 2017), a single neural network is trained $M$ times with the same data but different random seeds (thus with different initializations and data processing orders), yielding $M$ different models with parameters $\{\boldsymbol{\theta}_i\}_{i=1}^{M}$ located in different modes in the loss surface. For the prediction, the outputs from those $M$ models are averaged at the output level to construct a final output. The $M$ members participating in ensemble often disagree on their predictions, thus giving functional diversity facilitating more accurate, robust, and better calibrated decision makings. However, it requires $M$ number of models loaded on memory and $M$ number of forward computations.

The BN (Yun et al., 2023) is one of methods suggested for reducing computational costs of ensemble methods. BN hinges on the mode connectivity (Garipov et al., 2018) of ensemble members, meaning that it is possible to connect two different modes via a low-loss subspace, indicating the intrinsic connection between them. The main intuition behind BN is that, if we can learn such a low-loss subspace, then we may directly predict the outputs computed from the parameters on the subspace in the output level. In detail, consider a neural network $f_{\boldsymbol{\theta}}(\cdot)$ with a parameter $\boldsymbol{\theta}$ trained with a loss function $\mathcal{L}(\boldsymbol{\theta})$. Let $\boldsymbol{\theta}_i$ and $\boldsymbol{\theta}_j$ be two modes. BN first search for a new parameter $\boldsymbol{\theta}_{i,j}(\alpha)$ that

satisfies on the parametric Bezier curve,

$$\boldsymbol{\theta}_{i,j}(\alpha) = (1-\alpha)^2 \boldsymbol{\theta}_i + 2\alpha(1-\alpha)\boldsymbol{\theta}_{i,j} + \alpha^2 \boldsymbol{\theta}_j \tag{1}$$

where the anchor parameter $\boldsymbol{\theta}_{i,j}$ is obtained by minimizing $\mathbb{E}_{\alpha \sim \mathcal{U}(0,1)} [\mathcal{L}(\boldsymbol{\theta}_{i,j}(\alpha))]$, so that the parameters $\{\boldsymbol{\theta}_{i,j}(\alpha)\}_{\alpha \in [0,1]}$ on the curve locate on the low-loss subspace. After building such a curve, a light-weight neural network $s$ (BN) is trained, where $s$ gets a feature vector $\boldsymbol{z}_i$ of an input $\boldsymbol{x}$ computed solely from the first model $\boldsymbol{\theta}_i$ and try to predict the output evaluated at the center of the Bezier curve $\boldsymbol{\theta}_{i,j}(0.5)$ for the same input $\boldsymbol{x}$. That is, the BN is an estimator trying to approximate $f_{\boldsymbol{\theta}_{i,j}(0.5)}(\boldsymbol{x}) \approx s(\boldsymbol{z}_i)$. Then the ensemble of two models, $\boldsymbol{\theta}_i$ and $\boldsymbol{\theta}_{i,j}(0.5)$, can be approximated via the BN as follows:

$$\frac{1}{2}\left(f_{\boldsymbol{\theta}_i}(\boldsymbol{x}) + f_{\boldsymbol{\theta}_{i,j}(0.5)}(\boldsymbol{x})\right) \approx \frac{1}{2}\left(f_{\boldsymbol{\theta}_i}(\boldsymbol{x}) + s(\boldsymbol{z}_i)\right). \tag{2}$$

Since the $\boldsymbol{\theta}_{i,j}(0.5)$ encompasses the information of both $\boldsymbol{\theta}_i$ and $\boldsymbol{\theta}_j$, $s$ is expected to approximate the outputs to decent quality. However, since BN does not directly predict the output of $\boldsymbol{\theta}_j$ but only approximates the models with $\boldsymbol{\theta}_{i,j}(0.5)$, it has limitation in mimicking the actual ensemble predictions. In addition, a single bridge network is constructed between only a pair of given ensemble members. As a result, the number of BNs to be constructed may grow quadratically in the number of ensemble members, leading to extra training costs.

## 2.2 Diffusion Schrödinger Bridge

DSB (Bortoli et al., 2021; Chen et al., 2022) is a conditional diffusion model that solves Schrödinger Bridge (SB) problem (Schrödinger, 1932), an entropy-regularized optimal transport problem that finds the diffusion process between two distributions. Even though the SB problem provides the finite-time solution on finding the probability path, it requires iterative optimization and inference procedure which is time-consuming, and has rarely demonstrated its practicality in deep learning models albeit its theoretical soundness. Recently, Liu et al. (2023) proposed a tractable special case of DSB called Image-to-Image Schrödinger Bridge (I²SB) for an application of image manipulation such as image restoration or super-resolution. In their work, a tractable special case of DSB is proposed, and has demonstrated its efficiency for real-world image datasets. Due to its training stability incurred from training a single score network, we choose it out of several DSBs despite of its strict condition. In this section, we introduce a brief overview of I²SB method.

First of all, Schrödinger Bridge (SB) is an optimal transport problem that seeks to find the forward and backward processes

$$\mathrm{d}\boldsymbol{Z}_t = [f_t + \beta_t \nabla \log \Psi(\boldsymbol{Z}_t, t)]\mathrm{d}t + \sqrt{\beta_t}\mathrm{d}W_t, \quad \boldsymbol{Z}_0 \sim p_0$$
$$\mathrm{d}\boldsymbol{Z}_t = [f_t - \beta_t \nabla \log \hat{\Psi}(\boldsymbol{Z}_t, t)]\mathrm{d}t + \sqrt{\beta_t}\mathrm{d}\overline{W}_t, \quad \boldsymbol{Z}_1 \sim p_1 \tag{3}$$

where $(p_0, p_1)$ are the boundary distributions, $\{W_t, \overline{W}_t\}$ are the standard Wiener process and its time-reversal, and $\{f_t, \beta_t\}$ are the drift and diffusion coefficients. If the pair of functions $\{\Psi, \hat{\Psi}\}$ solves the following coupled PDE

$$\frac{\partial \Psi(\boldsymbol{z}_t, t)}{\partial t} = \nabla \Psi^\top f_t - \frac{1}{2}\beta_t \Delta \Psi, \quad \frac{\partial \hat{\Psi}(\boldsymbol{z}_t, t)}{\partial t} = -\nabla \cdot (\hat{\Psi} f_t) + \frac{1}{2}\beta_t \Delta \Psi \tag{4}$$
$$\text{with } \Psi(\boldsymbol{z}_0, 0)\hat{\Psi}(\boldsymbol{z}_0, 0) = p_0(\boldsymbol{z}_0), \Psi(\boldsymbol{z}_1, 1)\hat{\Psi}(\boldsymbol{z}_1, 1) = p_1(\boldsymbol{z}_1),$$

then (3) provides the optimal solution to an entropy-regularizing optimization problem that finds the optimal path between $p_0$ and $p_1$. Then (4) and its time-reversal directly follows the Fokker-Planck equation of the SDE in (5) as follows, respectively.

$$\mathrm{d}\boldsymbol{Z}_t = f_t\mathrm{d}t + \sqrt{\beta_t}\mathrm{d}W_t, \quad \boldsymbol{Z}_0 \sim \hat{\Psi}(\cdot, 0) \text{ and } \mathrm{d}\boldsymbol{Z}_t = f_t\mathrm{d}t + \sqrt{\beta_t}\mathrm{d}\overline{W}_t, \quad \boldsymbol{Z}_1 \sim \Psi(\cdot, 1). \tag{5}$$

However, both $\Psi$ and $\hat{\Psi}$ are intractable drifts so I²SB assumes a certain form of the boundary distributions $p_0$ and $p_1$. Followed from Liu et al. (2023), we take the energy potential functions $\widehat{\Psi}(\cdot, 0) = p_0(\cdot) := \delta_a(\cdot)$ and $\Psi(\cdot, 1) = p_1(\cdot)/\widehat{\Psi}(\cdot, 1)$. Here $\delta_a(\cdot)$ is the Dirac delta distribution centered at $a \in \mathbb{R}^d$ which makes the diffusion process computationally tractable. Then, we can approximate both forward and backward Stochastic Differential Equations (SDEs) using a single score network in the framework of DDPM (Ho et al., 2020) with the following Gaussian posterior:

$$\boldsymbol{Z}_t \sim q(\boldsymbol{Z}_t \mid \boldsymbol{Z}_0, \boldsymbol{Z}_1) = \mathcal{N}(\boldsymbol{Z}_t; \mu_t, \Sigma_t), \ \mu_t = \frac{\overline{\sigma}_t^2}{\overline{\sigma}_t^2 + \sigma_t^2}\boldsymbol{Z}_0 + \frac{\sigma_t^2}{\overline{\sigma}_t^2 + \sigma_t^2}\boldsymbol{Z}_1, \ \Sigma_t = \frac{\overline{\sigma}_t^2 \sigma_t^2}{\overline{\sigma}_t^2 + \sigma_t^2}, \tag{6}$$

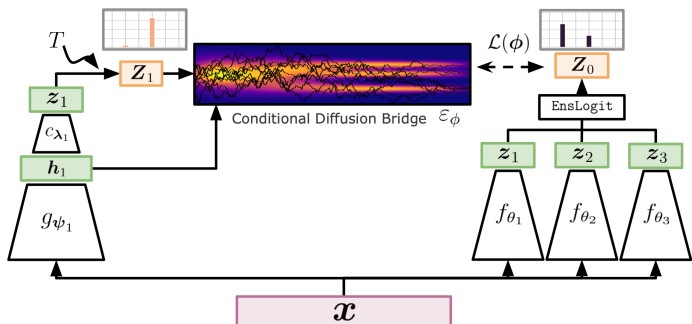

Figure 1: Overview of DBN. For a given data, the conditional diffusion bridge learns a transition between logit distribution of one of the ensembles (left; source) and that of the target ensemble models (right; target).

where $\sigma_t^2 = \int_0^t \beta_{t'} dt'$ and $\overline{\sigma}_t^2 = \int_t^1 \beta_{t'} dt'$ are cumulative forward and backward noise variances, respectively. For algorithmic design for restoration problems, we take $p(\boldsymbol{Z}_0, \boldsymbol{Z}_1) = p_0(\boldsymbol{Z}_0) p_1(\boldsymbol{Z}_1 | \boldsymbol{Z}_0)$ and $f = 0$, and construct tractable SBs between individual samples from $\boldsymbol{Z}_0$ and $p_1(\boldsymbol{Z}_1 | \boldsymbol{Z}_0)$.

## 3 DIFFUSION BRIDGE NETWORKS

In the previous BN (Yun et al., 2023), there was a limitation in learning the transport between independent models, prompting them to adopt an alternative approach: it utilizes a neural network to predict the output of local optima aligned between the source and target models in terms of a low loss subspace instead of directly predicting the output of the target model. To overcome this limitation, instead of tackling the problem of predicting the ensemble outputs as one-step prediction (a single neural network evaluation), we cast the problem as a diffusion bridge construction between the output distributions of ensemble members.

### 3.1 SETTINGS AND NOTATIONS

We restrict our focus on the $K$-way classification problem, where the goal is to train a classifier taking $d$-dimensional inputs and predict $K$-dimensional classification logits. We assume that a classifier with parameter $\boldsymbol{\theta}$ is decomposed into two parts, a feature extractor $g_{\boldsymbol{\psi}} : \mathbb{R}^d \to \mathbb{R}^h$ that first encodes an input $\boldsymbol{x}$ into a feature vector $\boldsymbol{h}$, and then a classifier $c_{\boldsymbol{\lambda}} : \mathbb{R}^h \to \mathbb{R}^K$ that transforms the feature vector $\boldsymbol{h}$ into a logit $\boldsymbol{z}$ of the class probability. That is, $f_{\boldsymbol{\theta}}(\boldsymbol{x}) = (c_{\boldsymbol{\lambda}} \circ g_{\boldsymbol{\psi}})(\boldsymbol{x})$ and $\boldsymbol{\theta} = (\boldsymbol{\psi}, \boldsymbol{\lambda})$. A collection of $M$ ensemble members are denoted as $\boldsymbol{\Theta} = \{\boldsymbol{\theta}_i\}_{i=1}^M$ and $\boldsymbol{\theta}_i = (\boldsymbol{\psi}_i, \boldsymbol{\lambda}_i)$.

### 3.2 CONDITIONAL DIFFUSION BRIDGE IN LOGIT SPACE

Our goal is to construct a conditional diffusion bridge on the logit space, that is, a stochastic path $\{\boldsymbol{Z}_t\}_{t \in [0,1]}$ where $\boldsymbol{Z}_1$ is constructed from the logit of the source model and $\boldsymbol{Z}_0$ from that of the target ensemble model. More specifically, given $\boldsymbol{\Theta}$, we choose one of the ensemble members $\boldsymbol{\theta}_1$ as a source model, and set $\boldsymbol{z}_1$ be the logit distribution computed from $\boldsymbol{\theta}_1$. Then the target $\boldsymbol{z}_0$ is set to be the logit distribution of the ensembled prediction.

More specifically, let $\boldsymbol{x} \in \mathbb{R}^d$ be an input, and let $\boldsymbol{h}_1 = g_{\boldsymbol{\psi}_1}(\boldsymbol{x})$ be the corresponding feature vector computed from the source model $\boldsymbol{\theta}_1$. Conditioned on $\boldsymbol{x}$, we define the source logit distribution as an implicitly defined distribution as follows,

$$\boldsymbol{z}_1 = c_{\boldsymbol{\lambda}_1}(\boldsymbol{h}_1), \quad T \sim p_{\text{temp}}, \quad \boldsymbol{Z}_1 = \frac{\boldsymbol{z}_1}{T}, \tag{7}$$

where $T$ is a annealing temperature drawn from some distribution $p_{\text{temp}}$. The target logit, $\boldsymbol{Z}_0$, is then set to be the logit computed by the ensemble model as follows, for $\boldsymbol{z}^{(i)} := f_{\boldsymbol{\theta}_i}(\boldsymbol{x})$,

$$\mathbf{p}_i = \texttt{Softmax}(\boldsymbol{z}^{(i)}), \quad \boldsymbol{Z}_0 = \texttt{EnsLogit}(\{\boldsymbol{z}^{(i)}\}_{i=1}^M) := \log \bar{\mathbf{p}} - \frac{1}{K} \sum_{k=1}^K \log \bar{\mathbf{p}}_{i,k}, \tag{8}$$

---

**Algorithm 1** Training DBNs

---

**Require:** An (empirical) data distribution $p_{\text{data}}$, a temperature distribution $p_{\text{temp}}$,
ensemble parameters $\{\boldsymbol{\theta}_i\}_{i=1}^M$, and the score network $\varepsilon_{\boldsymbol{\phi}}$.
Fix a source model $f_{\boldsymbol{\theta}_1}$.
**while** not converged **do**
    Sample $\boldsymbol{x} \sim p_{\text{data}}$.
    **for** $i = 1$ to $M$ **do**
        Compute the logits $\mathbf{z}_i = f_{\boldsymbol{\theta}_i}(\boldsymbol{x})$.
    **end for**
    Get a target ensemble logit $\mathbf{Z}_0 = \texttt{EnsLogit}(\{\boldsymbol{z}_i\}_{i=1}^M)$.
    Draw a temperarture $T \sim p_{\text{temp}}$ and compute the annealed source logit $\mathbf{Z}_1 = \boldsymbol{z}_1/T$.
    Compute the loss according to (10), and update $\boldsymbol{\phi} \leftarrow \boldsymbol{\phi} - \eta \nabla_{\boldsymbol{\phi}} \mathcal{L}(\boldsymbol{\phi})$.
**end while**
**return** $\boldsymbol{\phi}$.

---

where the ensemble output $\bar{\mathbf{p}} = \sum_{i=1}^M \mathbf{p}_i/M$. Then, we get $\texttt{Softmax}(\mathbf{Z}_0) = \bar{\mathbf{p}}$, allowing us to get ensemble output $\bar{\mathbf{p}}$ through approximating $\mathbf{Z}_0$. Next we construct $\text{I}^2\text{SB}$ between $\mathbf{Z}_1$ and $\mathbf{Z}_0$.

The intuition behind this construction is as follows. If we directly use the original logit $\boldsymbol{z}_1$, the learned DSB can easily be trapped in a trivial solution where it just produces the copy of the source logit $\boldsymbol{z}_1$ along the path $\{\mathbf{Z}_t\}_{t\in[0,1]}$, as the discrepancy between the source logit and the target logit is not large compared to the typical situation for which DSB is constructed. That is, the source $\boldsymbol{z}_1$ can be a strong hint that acts a simplicity bias for DSB learning. Also, $\text{I}^2\text{SB}$ requires the source of the diffusion to be a conditional probability distribution, but $\boldsymbol{z}_1$ is a deterministic value. Hence by randomly annealing the source logit via a temperature $T$, we can naturally construct the source as the distribution of the annealed logits and also dilute the information included in $\boldsymbol{z}_1$, and this encourages DSB to discover non-trivial paths between the source and the target.

Based on the formulation on constructing the conditional diffusion bridge between the source and target distributions, we build an approximate reverse SDE derived from (3) that simulates the path from $\mathbf{Z}_1$ to $\mathbf{Z}_0$ by estimating the score function $\nabla \log \hat{\Psi}(\mathbf{Z}_t, t|\boldsymbol{h}_1) = \varepsilon_{\boldsymbol{\phi}}(\boldsymbol{h}_1, \mathbf{Z}_t, t)/\beta_t$ as

$$\text{d}\mathbf{Z}_t = (\beta_t/\sigma_t)\varepsilon_{\boldsymbol{\phi}}(\boldsymbol{h}_1, \mathbf{Z}_t, t)\text{d}t + \sqrt{\beta_t}\text{d}W_t, \quad \mathbf{Z}_1 \sim p_1(\mathbf{Z}_1 \,|\, \boldsymbol{x}), \tag{9}$$

where $p_1$ is the distribution of $\mathbf{Z}_1$ implicitly defined as in (7) and $\varepsilon_{\boldsymbol{\phi}} : \mathbb{R}^h \times \mathbb{R}^K \times [0, 1] \to \mathbb{R}^K$ is the score function estimator built with a neural network parameterized by $\boldsymbol{\phi}$, and is trained to estimate the score function $\nabla_{\mathbf{Z}} \log p_t(\boldsymbol{z}_t \,|\, \boldsymbol{x})$ by minimizing the following objective function,

$$\mathcal{L}(\boldsymbol{\phi}) = \mathbb{E}_{\boldsymbol{x}, \mathbf{Z}_t} \left[ \left\| \varepsilon_{\boldsymbol{\phi}}(\boldsymbol{h}_1, \mathbf{Z}_t, t) - \frac{\mathbf{Z}_t - \mathbf{Z}_0}{\sigma_t} \right\|_2^2 \right], \tag{10}$$

with $\boldsymbol{x} \sim p_{\text{data}}$, $t \in \mathcal{U}([0, 1])$, and $\mathbf{Z}_t \sim q(\mathbf{Z}_t \,|\, \mathbf{Z}_0, \mathbf{Z}_1)$ as defined in (6). The overall training pipeline is summarized in Algorithm 1.

### 3.3 DISTILLATION OF DIFFUSION BRIDGE

Even though the family of diffusion models (Ho et al., 2020; Bortoli et al., 2021; Liu et al., 2023) achieves superior performance in learning generative models or constructing paths between distributions, they typically suffer from the slow sampling speed due to a large number of function evaluations required for simulation. The distillation techniques for diffusion models, which distill multiple steps of the reverse diffusion process to a single step, do not significantly harm the generation performance while accelerating the sampling speed. In this paper, we adapt the progressive distillation proposed in (Salimans & Ho, 2022) to reduce the sampling cost of DBN.

Let $\mathcal{T} = \{t_i\}_{i=1}^N$ be the discretized time interval used for diffusion bridge, with $0 = t_0 < t_1 < \cdots < t_N = 1$. Then let $\mathcal{T}' := \{t'_j\}_{j=1}^{N'}$ be the distilled time interval with $\mathcal{T}' \subset \mathcal{T}$. Let $\mathbf{Z}_{t_{i-1}} \sim p_{\boldsymbol{\phi}}(\mathbf{Z}_{t_{i-1}} \,|\, \mathbf{Z}_{t_i})$ be the sample from an ancestral sampling following the score network $\varepsilon_{\boldsymbol{\phi}}$. Then a distilled score network with parameter $\boldsymbol{\phi}'$ is then trained with the loss

$$\mathcal{L}_{\text{distill}}(\boldsymbol{\phi}') = \mathbb{E}_{\boldsymbol{x}, j} \left[ \left\| \varepsilon_{\boldsymbol{\phi}'}(\boldsymbol{h}_1, \mathbf{Z}_{t'_j}, t'_j) - \frac{\mathbf{Z}_{t'_j} - \mathbf{Z}_{t'_{j-1}}}{\sigma_{t'_j}} \right\|_2^2 \right], \tag{11}$$

where $j \sim \mathcal{U}(\{1, \ldots, N'\})$ and $\boldsymbol{Z}_{t'_{j-1}} \sim \prod_{s=i-k}^{i-1} p_{\phi}(\boldsymbol{Z}_{t_s} \mid \boldsymbol{Z}_{t_{s+1}})$ with $t'_{j-1} = t_{i-k}$ and $t'_j = t_i$. This distillation is then recursively repeated until there only remains a single time step ($N' = 1$).

### 3.4 INFERENCE PROCEDURE

The inference with DBN consists of forwarding an input through the source model and computing a single diffusion step from the model distilled by (11). Given an input $\boldsymbol{x}$, we first compute its feature $\boldsymbol{h}_1$ using the feature extractor of the source model $g_{\boldsymbol{\psi}_1}$ with $\boldsymbol{h}_1 = g_{\boldsymbol{\psi}_1}(\boldsymbol{x})$. Then we initialize the diffusion by drawing $T \sim p_{\text{temp}}$ and put $\boldsymbol{Z}_1 = \boldsymbol{z}_1/T$. The corresponding ensembled prediction $\boldsymbol{y}$ is then approximated as,

$$\boldsymbol{Z}_0 = \boldsymbol{Z}_1 + (\beta_1/\sigma_1)\varepsilon_{\phi'}(\boldsymbol{h}_1, \boldsymbol{Z}_1, 0) + \boldsymbol{\xi}_1 \text{ where } \boldsymbol{\xi}_1 \sim \mathcal{N}(0, \Sigma_1),$$

$$p(\boldsymbol{y} \mid \boldsymbol{x}, \{\boldsymbol{\theta}_i\}_{i=1}^M) = \frac{1}{M}\sum_{i=1}^M \texttt{Softmax}(\boldsymbol{z}^{(i)}) \approx \texttt{Softmax}(\boldsymbol{Z}_0). \tag{12}$$

### 3.5 COMBINING MULTIPLE DBNS

Note that the size of the score network $\varepsilon_{\phi'}$ should be limited, because otherwise the cost from the diffusion simulation can outnumber the cost of computing the full ensemble. Hence, there is an intrinsic limit in the capacity of $\varepsilon_{\phi'}$ representing the path between the source model and an full ensembled model (we study this capacity empirically in § 4.3). When a single DBN reached out to the limit, we may introduce multiple DBNs, increasing the approximation quality at the cost of additional inference time. Given $M$ ensemble models, we first build a DBN starting from a source model $\boldsymbol{\theta}_1$. Then we build another DBN sharing the same model $\boldsymbol{\theta}_1$ as the source and so on. Let $\{\varepsilon_{\phi'}^{(\ell)}\}_{\ell=1}^L$ be a set of score networks built in that way, with $L < M$. Then we can approximate the ensembled prediction $\boldsymbol{y}$ for $\boldsymbol{x}$ as,

$$p(\boldsymbol{y} \mid \boldsymbol{x}, \{\boldsymbol{\theta}_i\}_{i=1}^M) \approx \frac{1}{L}\sum_{\ell=1}^L \texttt{Softmax}\left(\boldsymbol{Z}_0^{(\ell)}\right), \tag{13}$$

where $\boldsymbol{Z}_0^{(\ell)}$ is the sample drawn as in (12) with $\ell^{\text{th}}$ score network. In our implementation, we combine $L$ DBNs where $l^{\text{th}}$ DBN estimates the recovered predictions by a collection of $N + 1$ models $\{f_{\boldsymbol{\theta}_1}, f_{\boldsymbol{\theta}_{(l-1)(N-1)+2}}, \cdots, f_{\boldsymbol{\theta}_{(l-1)(N-1)+N+1}}\}$, approximating the ensembled prediction of $M = LN + 1$ ensemble models. For example, to simulate total $M = 5$ ensemble models $\{\boldsymbol{\theta}_1, \boldsymbol{\theta}_2, \boldsymbol{\theta}_3, \boldsymbol{\theta}_4, \boldsymbol{\theta}_5\}$, we first train a score net that estimate ensemble distribution of $\{\boldsymbol{\theta}_1, \boldsymbol{\theta}_2, \boldsymbol{\theta}_3\}$ starting from $\boldsymbol{\theta}_1$ and secondly train another score net approximating $\{\boldsymbol{\theta}_1, \boldsymbol{\theta}_4, \boldsymbol{\theta}_5\}$ starting from $\boldsymbol{\theta}_1$. Note that the source models for $L$ DBNs need not be different; in § 4, we show that a *single* source model can be shared for all multiple DBNs, minimizing the additional inference cost while significantly improving the accuracy.

## 4 EXPERIMENTS

**Settings.** We evaluate our approach using three widely adopted image classification benchmark datasets: CIFAR-10, CIFAR-100, and TinyImageNet (Li et al., 2017). In our experiment, the ensemble models that we construct to serve as bridges adopt the configuration outlined in the Bridge Network (Yun et al., 2023) and are trained based on the ResNet architecture. We use widely used ResNet-32×2, ResNet-32×4, and ResNet-34 networks (He et al., 2016) for baseline ensemble classifiers for CIFAR-10, CIFAR-100, and TinyImageNet datasets, respectively. In this context, the suffices "×2" and "×4" denote that the channel widths of the convolutional layers are multiplied by 2 and 4 from the conventional ResNet-32, respectively. For the score network, inspired by Sandler et al. (2018), we utilize residual connections (He et al., 2016) and Depthwise Separable Convolution (DSC) (Chollet, 2017) to mitigate redundant computations and implement a light-weighted score network. Further details on datasets, model architectures, and hyperparameter settings used to evaluate our experiments are listed in Appendix D.

**Training.** We train a single score network with 3 ensembles. If we need to mimic more than 4 ensembles, we can average 2 or more diffusion bridges according to § 3.5. Since they share the

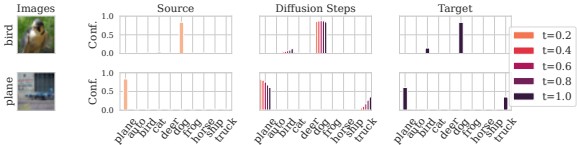

Figure 2: Confidences from the source model (first column), from the ensemble model (third column), and from the diffusion bridge (middle column) in the CIFAR-10 dataset. The middle column illustrates a transition of the diffusion process.

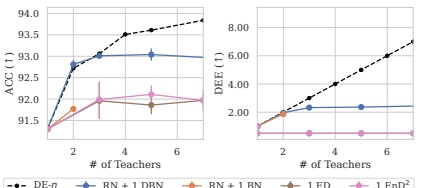

Figure 3: The number of teachers that a single model can distill in terms of ACC (left) and DEE (right). DEEs less than 1 are set to 0.5.

source model, we can easily enhance the performance with low extra costs. We train the diffusion bridge with 5 steps before distillation for fast and efficient training of the score networks and 5 steps are enough to approximate the transport between the two conditional logit distributions.

**Baseline methods.** We validate how well our method accelerates the inference speed with lightweight networks by comparing our DBN with the target DE (Lakshminarayanan et al., 2017) as the oracle ensemble. We also compare our method with three existing methods that are widely used in fast and efficient ensemble distillation or bridge networks: Ensemble Distillation (ED) (Hinton et al., 2015), Ensemble Distribution Distillation (EnD$^2$) (Ryabinin et al., 2021), and BN (Yun et al., 2023). We use more refined version of EnD$^2$ (Ryabinin et al., 2021) instead of the original EnD$^2$ (Malinin et al., 2020) that improved convergence.

**Metrics.** We measure the computational cost of each model in terms of FLoating point OPerations (FLOPs) and the number of parameters (#Params). FLOPs count the number of additions and multiplications operations, and it represents a cost during an inference of a model. #Params represents the memory usage required for the inference. For the performance, we consider the classification accuracy (ACC), Negative Log-Likelihood (NLL), Brier Score (BS) (Brier, 1950), Expected Calibration Error (ECE) (Guo et al., 2017), and Deep Ensemble Equivalent (DEE) (Ashukha et al., 2020). BS and ECE measure how much the classification output (confidence) is aligned with the true probability and thereby it implies the reliability of the model output. DEE approximates the number of DE similar to a given model in terms of NLL. More profound formulations of the metrics are described in Appendix A.3.

## 4.1 CLASSIFICATION PERFORMANCE AND UNCERTAINTY METRICS

The ACC, NLL, BS, ECE, and DEE comparisons with the baselines with respect to FLOPs and #Params on CIFAR-10, CIFAR-100 and TinyImageNet are shown in Table 1. We assume the situation where both BN and DBN can utilize only a single source model to make the problem difficult. As we can see in the results of CIFAR-10 and CIFAR-100, with only a small increase in the computational costs (FLOPs and #Params), DBN achieves almost DE-3 performance, whereas BN struggles to achieve even DE-2 performance with more computational costs than DBN. In TinyImageNet, DBN even outperforms DE-3 with less than a half of computation costs. On the other hand, the two distillation methods, ED and EnD$^2$, are competitive with BN but they shows poor uncertainty metrics such as NLL, BS, and ECE, and interestingly DBN also shows poor ECE scores even with high performance in the other uncertainty metrics. In addition, the actual output results through the diffusion processes are illustrated in Figure 2 and the results for the other images are listed in Appendix C.

## 4.2 PERFORMANCE LOSS VS. COMPUTATIONAL EFFICIENCY

Furthermore, depending on the number of ensemble models that DBN is trained on, the left hand side of Figure 4 demonstrates how much performance loss occurs compared to the DE when the number of target ensemble models increase. Conversely, the right hand side of Figure 4 illustrates how much cost savings DBN can achieve compared to DE, given the same computational cost (FLOPs). The comparison is made from the perspectives of ACC and DEE. In this experiment, a maximum of 9 ensemble models are used and we also compare with our competing model, BN. DBN trains one diffusion bridge with three ensembles, while BN learns a low-loss curve between 2 ensembles for

Table 1: Performance on CIFAR-10/100, and TinyImageNet. $\text{BN}_{medium}$ is the standard size of BN and $\text{BN}_{small}$ has reduced channels compared to $\text{BN}_{medium}$. The parentheses next to each model (e.g. (DE-3)) means the number of DEs that each model learns as a target.

**CIFAR-10**

| Model | FLOPs (↓) | #Params (↓) | ACC (↑) | NLL (↓) | BS (↓) | ECE (↓) | DEE (↑) |
|---|---|---|---|---|---|---|---|
| ResNet (DE-1) | × 1.000 | × 1.000 | 91.30 ± 00.10 | 0.3382 ± 0.0023 | 0.1409 ± 0.0011 | 0.0658 ± 0.0003 | 1.000 |
| +2 $\text{BN}_{small}$ (DE-3) | × 1.125 | × 1.097 | 91.79 ± 00.05 | 0.2579 ± 0.0009 | 0.1198 ± 0.0003 | 0.0599 ± 0.0037 | 1.899 |
| +4 $\text{BN}_{small}$ (DE-5) | × 1.245 | × 1.283 | 91.87 ± 00.05 | 0.2580 ± 0.0010 | 0.1195 ± 0.0004 | 0.0624 ± 0.0098 | 1.898 |
| +2 $\text{BN}_{medium}$ (DE-3) | × 1.411 | × 1.319 | 91.91 ± 00.04 | 0.2544 ± 0.0011 | 0.1182 ± 0.0005 | **0.0591** ± 0.0096 | 1.938 |
| +1 DBN (DE-3) | × 1.166 | × 1.213 | 92.98 ± 00.16 | **0.2403** ± 0.0027 | **0.1084** ± 0.0013 | 0.0666 ± 0.0010 | **2.363** |
| +2 DBN (DE-5) | × 1.332 | × 1.426 | **93.23** ± 00.07 | **0.2247** ± 0.0005 | **0.1033** ± 0.0005 | 0.0662 ± 0.0009 | **3.031** |
| ED (DE-3) | × 1.000 | × 1.000 | 91.96 ± 00.42 | 0.3505 ± 0.0214 | 0.1366 ± 0.0074 | 0.0674 ± 0.0037 | <1 |
| ED (DE-5) | × 1.000 | × 1.000 | 91.86 ± 00.21 | 0.3577 ± 0.0083 | 0.1391 ± 0.0031 | 0.0683 ± 0.0017 | <1 |
| $\text{EnD}^2$ (DE-3) | × 1.000 | × 1.000 | 91.99 ± 00.14 | 0.3405 ± 0.0079 | 0.1358 ± 0.0026 | 0.0690 ± 0.0013 | <1 |
| $\text{EnD}^2$ (DE-5) | × 1.000 | × 1.000 | 92.11 ± 00.23 | 0.3313 ± 0.0057 | 0.1336 ± 0.0029 | 0.0645 ± 0.0014 | 1.077 |
| DE-2 | × 2.000 | × 2.000 | 92.72 ± 00.13 | 0.2489 ± 0.0031 | 0.1125 ± 0.0012 | **0.0484** ± 0.0010 | 2.000 |
| DE-3 | × 3.000 | × 3.000 | 93.06 ± 00.14 | 0.2252 ± 0.0024 | 0.1038 ± 0.0008 | 0.0469 ± 0.0012 | 3.000 |
| DE-5 | × 5.000 | × 5.000 | **93.61** ± 00.11 | **0.2005** ± 0.0015 | **0.0951** ± 0.0004 | 0.0466 ± 0.0009 | 5.000 |

**CIFAR-100**

| Model | FLOPs (↓) | #Params (↓) | ACC (↑) | NLL (↓) | BS (↓) | ECE (↓) | DEE (↑) |
|---|---|---|---|---|---|---|---|
| ResNet (DE-1) | × 1.000 | × 1.000 | 72.29 ± 00.36 | 1.1506 ± 0.0100 | 0.4001 ± 0.0044 | 0.1526 ± 0.0005 | 1.000 |
| +2 $\text{BN}_{medium}$ (DE-3) | × 1.419 | × 1.320 | 74.97 ± 00.05 | 1.0360 ± 0.0022 | 0.3500 ± 0.0007 | **0.1228** ± 0.0210 | 1.642 |
| +1 DBN (DE-3) | × 1.166 | × 1.213 | 76.02 ± 00.11 | **0.9434** ± 0.0051 | **0.3438** ± 0.0009 | 0.1352 ± 0.0020 | **2.461** |
| +2 DBN (DE-5) | × 1.332 | × 1.426 | 76.82 ± 00.22 | **0.8998** ± 0.0046 | **0.3305** ± 0.0018 | 0.1269 ± 0.0013 | **3.297** |
| ED (DE-3) | × 1.000 | × 1.000 | 74.18 ± 00.22 | 1.2375 ± 0.0125 | 0.4095 ± 0.0019 | 0.1819 ± 0.0013 | <1 |
| ED (DE-5) | × 1.000 | × 1.000 | 74.00 ± 00.29 | 1.2428 ± 0.0174 | 0.4120 ± 0.0041 | 0.1840 ± 0.0022 | <1 |
| $\text{EnD}^2$ (DE-3) | × 1.000 | × 1.000 | 73.35 ± 00.22 | 1.3572 ± 0.0079 | 0.4350 ± 0.0013 | 0.1973 ± 0.0006 | <1 |
| $\text{EnD}^2$ (DE-5) | × 1.000 | × 1.000 | 73.22 ± 00.33 | 1.3597 ± 0.0156 | 0.4370 ± 0.0052 | 0.1980 ± 0.0030 | <1 |
| DE-2 | × 2.000 | × 2.000 | 74.98 ± 00.42 | 0.9721 ± 0.0099 | 0.3505 ± 0.0034 | **0.1259** ± 0.0021 | 2.000 |
| DE-3 | × 3.000 | × 3.000 | **76.04** ± 00.13 | **0.9098** ± 0.0019 | **0.3342** ± 0.0007 | **0.1233** ± 0.0020 | **3.000** |
| DE-5 | × 5.000 | × 5.000 | 77.03 ± 00.08 | **0.8606** ± 0.0036 | **0.3216** ± 0.0013 | **0.1234** ± 0.0019 | **5.000** |

**TinyImageNet**

| Model | FLOPs (↓) | #Params (↓) | ACC (↑) | NLL (↓) | BS (↓) | ECE (↓) | DEE (↑) |
|---|---|---|---|---|---|---|---|
| ResNet (DE-1) | × 1.000 | × 1.000 | 59.28 ± 00.46 | 1.8356 ± 0.0186 | 0.5426 ± 0.0052 | 0.2024 ± 0.0025 | 1.000 |
| +2 $\text{BN}_{medium}$ (DE-3) | × 1.359 | × 1.412 | 64.07 ± 0.16 | **1.5066** ± 0.0021 | 0.4745 ± 0.0007 | **0.1267** ± 0.0137 | **4.102** |
| +1 DBN (DE-3) | × 1.209 | × 1.149 | **64.60** ± 00.06 | 1.5299 ± 0.0022 | **0.4718** ± 0.0013 | 0.1836 ± 0.0021 | 3.444 |
| +2 DBN (DE-5) | × 1.418 | × 1.298 | **65.34** ± 00.10 | **1.4890** ± 0.0045 | **0.4616** ± 0.0016 | 0.1803 ± 0.0019 | **4.701** |
| ED (DE-3) | × 1.000 | × 1.000 | 60.81 ± 00.26 | 1.8312 ± 0.0129 | 0.5443 ± 0.0052 | 0.2126 ± 0.0021 | <1 |
| $\text{EnD}^2$ (DE-3) | × 1.000 | × 1.000 | 60.71 ± 00.31 | 1.9991 ± 0.0165 | 0.5828 ± 0.0026 | 0.2279 ± 0.0010 | <1 |
| DE-2 | × 2.000 | × 2.000 | 62.47 ± 00.35 | 1.6264 ± 0.0110 | 0.4942 ± 0.0033 | 0.1834 ± 0.0013 | 2.000 |
| DE-3 | × 3.000 | × 3.000 | 63.78 ± 00.23 | 1.5538 ± 0.0055 | 0.4788 ± 0.0015 | 0.1836 ± 0.0021 | 3.000 |

one bridge. For more than four ensembles, DBN conducts ensemble inference aggregating two or more diffusion bridges as BN does. We use the standard size of BN in "ResNet + 2 BNs" which has 1.411 relative FLOPs for a single network compared to "ResNet + 1 DBNs" which has 1.166 relative FLOPs. As shown in Figure 4, DBN (left) achieves significant ensemble gains even when the number of target ensemble increases, whereas BN (left) saturates at ACC 92.0% and DEE 2%. Moreover, DBN (right) shows rapid ensemble inference compared to the target ensemble DE (right) and even faster than BN (right) to get the same ensemble performance.

### 4.3 Ensemble Capacity of a Single DBN

Figure 3 demonstrates the ensemble capacity of how much a single DBN model learns and distills knowledge from multiple teacher (ensemble) models in terms of ACC and DEE in the CIFAR-10 dataset. Existing ensemble distillation models have limited performance in learning likelihood and are saturated in terms of ACC at learning at most 3 number of target ensembles. On the other hand, BN assumes that a single model can cover at most two ensemble modes, hence the maximum gain on ACC and DEE is capped at two. Compared to these competing methods, our DBN method succeeds in learning slightly less than three ensembles only with a single lightweight model and the source model.

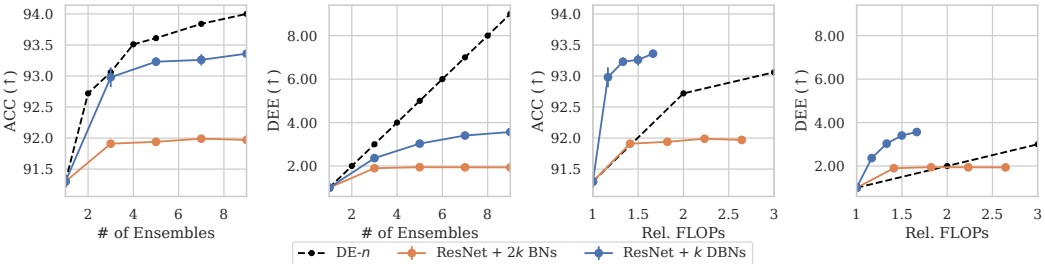

Figure 4: From **left** (1st) to **right** (4th): Number of target ensembles vs. accuracy (1st) or DEE (2nd), and relative FLOPs vs. accuracy (3rd) or DEE (4th) in CIFAR-10.

## 5  RELATED WORK

*Fast Ensembling Methods for Neural Network Prediction.* Many works have suggested to reduce the computational cost of the typical ensemble methods that are multiplied proportional to the number of the ensemble networks. To provide output diversity with lightweight computation, sharing either weights or latent features of neural networks enabled to compress redundancy throughout the ensemble members and add minor differences between them (Wen et al., 2019; Dusenberry et al., 2020; Lee et al., 2015; Siqueira et al., 2018; Antorán et al., 2020; Havasi et al., 2021). This can also be interpreted as the knowledge distillation from the ensemble of networks to the single network; (Hinton et al., 2015), Malinin et al. (2020); Ryabinin et al. (2021); Penso et al. (2022) showed that distilling deep ensemble to a single network helps retaining information of the ensemble distribution without heavy computational burden.

*Restoration with Conditional Diffusion Models.* Our model is on the line of the reconstruction problem from some degraded measurement, if we consider our method as reconstructing the ensemble distribution from the degraded function output from a single model. This line of work starts with the conditional diffusion model (Saharia et al., 2022), that refines the image given some conditional features. Conditional diffusion models have achieved success in various problems such as time series imputation (Tashiro et al., 2021), deblurring (Whang et al., 2022), and super-resolution (Saharia et al., 2023). As a generalized perspective, inverse problems dealt with diffusion (Song et al., 2022) aims to restore the underlying clean signal from the noisy measurement. Wang et al. (2023) delves into the null space of the image and utilize the valid space that should be recovered, and achieved zero-shot restoration using diffusion models.

## 6  CONCLUSION

We have proposed a novel approach to approximate the performance of Deep Ensembles with reduced computational cost. To achieve this, we constructed Conditional Diffusion Bridge that connects the logit distribution between one of the ensemble models and the target ensemble. We approximated the output distribution of the target ensemble using the features and logits obtained from the source model. The computations involved in this process consist of a single forward pass of the source model and the diffusion process with a lightweight score network. Additionally, during the training process, distilling the multiple diffusion steps into one step accelerates the inference speed while retaining the performance of three deep ensemble models. We evaluated this method on three widely used datasets and achieved superior performance compared to the baselines. Notably, we demonstrated significantly faster computations compared to competitive models such as Bridge Network while achieving similar performance. However, there are still some limitations to consider. First, a single Diffusion Bridge still has limitations in terms of the number of ensembles it can learn. If more ensembles are required, additional diffusion bridges must be used. Secondly, the use of multiple diffusion bridges leads to a proportional training time because diffusion models demand a long training time due to their multiple diffusion steps.

ACKNOWLEDGMENTS

This work is partly supported by Institute for Information & communications Technology Planning & Evaluation (IITP) grant funded by the Korea government (MSIT) (No.2019-0-00075: Artificial Intelligence Graduate School Program (KAIST), No.2022-0-00713: Meta-learning Applicable to Real-world Problems, No.2022-0-00184: Development and Study of AI Technologies to Inexpensively Conform to Evolving Policy on Ethics), and KAIST-NAVER Hypercreative AI Center. Our research is supported with Cloud TPUs from Google's TPU Research Cloud (TRC).

**Ethics statement.** This paper does not include any ethical issues. This paper presents a fast ensemble inference algorithm of mainly image classifications which does not cause ethical issues.

**Reproducibility statement.** We described our experimental details in Appendix D and Appendix A.3 which includes information about datasets, architectures, and hyperparameters used.

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

## A    EXPERIMENTAL DETAILS

We describe the overall details of our experiment below.

### A.1    DATASETS

We employ CIFAR-10/100 (Krizhevsky et al., 2009), and TinyImageNet (Li et al., 2017) datasets for our study. Our data augmentation strategy involves randomly cropping images by 32 pixels with an additional 4-pixel padding, as well as applying random horizontal flipping. Furthermore, we normalize input images by subtracting per-channel means and dividing them by per-channel standard deviations.

### A.2    TARGET MODEL ARCHITECTURES

In our investigation, we implement comparable ResNet block configurations. The overall network architectures are consistent with one of our baseline, BN (Yun et al., 2023), to compare in a reliable condition with a minor difference in TinyImageNet and ImageNet.

**Classifier Architectures.**

**CIFAR-10.**    ResNet-32 × 2, characterized by 15 basic blocks distributed as (5, 5, 5) and a total of 32 layers with the Filter Response normalization (FRN) (Singh & Krishnan, 2020) and the Swish activation layer. This model incorporates a widen factor of 2 and operates with in-planes set at 16.

**CIFAR-100.**    ResNet-32 × 4, which closely resembles the CIFAR-10 network with a widen factor of 4 with FRN and the Swish activation.

**TinyImageNet.**    ResNet-34, which encompasses 16 basic blocks organized as (3, 4, 6, 3) and 34 layers in total with FRN and Swish. The in-planes parameter for this model is established at 64. The only difference with the target model of Yun et al. (2023) is that we use bias in the convolutional layers but they don't.

**ImageNet64.**    ResNet-34, which encompasses 16 basic blocks organized as (3, 4, 6, 3) and 34 layers in total with FRN and Swish. The in-planes parameter for this model is established at 64 as in TinyImageNet. However, we don't use bias in every convolutional layers in this task.

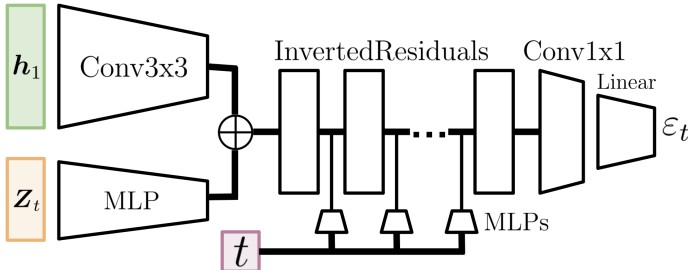

Figure 5: Score network architecture.

**Score Network Architectures.**    The score networks consist of three embedding networks for $h_1$, $Z_t$, and $t$, some inverted residual blocks (Sandler et al., 2018), and the output layer. The embedding network for $Z_t$ consists of a LayerNorm layer followed by the average pooling layer, and returns the embedding output by a forward computation to a lightweight MLP (width $2 \rightarrow 2 \rightarrow 1$ with the ReLU6 activation layer), and concatenate the embedding outputs to the embeddings of $h_1$. The embedding network for $t$ firstly use the sinusoidal time-step embeddings with the maximum period 10,000 and then feed-forward to a light weight MLP (width $d/4 \rightarrow d/2 \rightarrow e$ with the Swish activation, where $d$ is the dimension of the logits and $e$ is the input channels of each inverted residual blocks.) The specifications of the inverted residuals and the size of the output layers in the score networks vary across the tasks as in Table 2.

Table 2: Specification of the inverted residual blocks and output layers in the score network across the tasks. $t$ is the expansion factor (Sandler et al., 2018). $c$ is the output channels. $n$ is the number of layers. $s$ is the stride applied in the first layer of each inverted residual.

| CIFAR10 | | | | | CIFAR100 | | | | | TinyImageNet | | | | | ImageNet64 | | | | |
|---|---|---|---|---|---|---|---|---|---|---|---|---|---|---|---|---|---|---|---|
| Operator | $t$ | $c$ | $n$ | $s$ | Operator | $t$ | $c$ | $n$ | $s$ | Operator | $t$ | $c$ | $n$ | $s$ | Operator | $t$ | $c$ | $n$ | $s$ |
| InvertedResidual | 1 | 32 | 1 | 1 | InvertedResidual | 1 | 64 | 1 | 1 | InvertedResidual | 1 | 64 | 1 | 1 | InvertedResidual | 6 | 64 | 3 | 2 |
| InvertedResidual | 3 | 48 | 2 | 2 | InvertedResidual | 3 | 96 | 2 | 2 | InvertedResidual | 4 | 96 | 2 | 2 | InvertedResidual | 6 | 96 | 3 | 2 |
| InvertedResidual | 3 | 64 | 3 | 2 | InvertedResidual | 3 | 128 | 3 | 2 | InvertedResidual | 4 | 128 | 3 | 2 | InvertedResidual | 6 | 160 | 3 | 1 |
| InvertedResidual | 3 | 96 | 2 | 1 | InvertedResidual | 3 | 192 | 2 | 1 | InvertedResidual | 4 | 192 | 4 | 1 | InvertedResidual | 6 | 320 | 3 | 2 |
| InvertedResidual | 3 | 128 | 2 | 1 | InvertedResidual | 3 | 256 | 2 | 1 | InvertedResidual | 4 | 256 | 3 | 2 | InvertedResidual | 6 | 640 | 1 | 1 |
| Conv 1x1 | - | 128 | 1 | 1 | Conv 1x1 | - | 256 | 1 | 1 | Conv 1x1 | - | 256 | 1 | 1 | Conv 1x1 | - | 1280 | 1 | 1 |
| AvgPool | - | - | 1 | - | AvgPool | - | - | 1 | - | AvgPool | - | - | 1 | - | AvgPool | - | - | 1 | - |
| Linear | - | 10 | 1 | - | Linear | - | 100 | 1 | - | Linear | - | 200 | 1 | - | Linear | - | 1000 | 1 | - |

## A.3 METRICS

We introduce the metrics used in our experiments.

- Accuracy

$$\mathbb{E}_{(\boldsymbol{x},y)}\Big[I[y = \arg\max_{k} \boldsymbol{p}^{(k)}(\boldsymbol{x})]\Big], \tag{14}$$

  where $I$ is the indicator function.

- Negative log-likelihood (NLL)

$$\mathbb{E}_{(\boldsymbol{x},y)}\Big[-\log \boldsymbol{p}^{(y)}(\boldsymbol{x})\Big]. \tag{15}$$

- Brier score (BS)

$$\mathbb{E}_{(\boldsymbol{x},y)}\Big[\big\|\boldsymbol{p}(\boldsymbol{x}) - \boldsymbol{y}\big\|_2^2\Big], \tag{16}$$

  where $\boldsymbol{y}$ is a one-hot-encoded label $y$.

- Expected calibration error (ECE)

$$\mathrm{ECE}(\mathcal{D}, N_{\mathrm{bin}}) = \sum_{b=1}^{N_{\mathrm{bin}}} \frac{n_b|\delta_b|}{n_1 + \cdots + n_{N_{\mathrm{bin}}}}, \tag{17}$$

  where $N_{\mathrm{bin}}$ is the quantity of bins, $n_b$ is the number of instances within the $b^{\mathrm{th}}$ bin, and $\delta_b$ is the calibration discrepancy associated with the $b^{\mathrm{th}}$ bin. To elaborate, the $b^{\mathrm{th}}$ bin encompasses predictions characterized by the highest confidence levels falling within the interval $[(b-1)/K, b/K)$, and the calibration error is defined as the disparity between accuracy and the mean confidence values. We maintains $N_{\mathrm{bin}} = 15$ throughout the paper.

- Deep ensemble equivalent score (DEE)

  First introduced in Ashukha et al. (2020), this metric assumes that the NLL of the DE-$k$ model decreases monotonely by $k$, and obtain how equivalent the objective model to how many ensemble of the baseline model, as

$$\mathrm{DEE}(f) = s + \frac{\mathrm{NLL}(f) - \mathrm{NLL}(f_s)}{\mathrm{NLL}(f_{s+1}) - \mathrm{NLL}(f_s)},$$
$$s = \arg\max_{i}\{i \in \mathbb{N} : \mathrm{NLL}(f_i) \geq \mathrm{NLL}(f)\} \tag{18}$$

  where $f_i$ is the DE-$i$ model. In our paper, DEE is linearly extrapolated below 1 if $\mathrm{NLL}(f) > \mathrm{NLL}(f_1)$.

## B HYPERPARAMETER SETTINGS

We list common hyperparameters for training DBNs in every dataset as follows:

**Optimizer.** For training our DBN model, we use the ADAM (Kingma & Ba, 2015) optimizer with zero weight decay and apply consine-decay as a learning-rate scheduling. For training the teacher ensemble model, we followed the BN paper by using the SGD optimizer with weight decay, as followed in Table 4.

**Regularization.** We use Exponential Moving Average (EMA) with 0.99995 decay factor and the mixup augmentation with $\alpha = 0.4$ except for ImageNet, following (Yun et al., 2023).

**Diffusion Model.** Our DBNs follow the training policy of discrete-time conditional diffusion model with uniform timesteps. After training the baseline DBNs, we distill them to one step, following Salimans & Ho (2022).

**Feature vector $h_1$.** We exploit the output of the first residual block of the teacher ResNet.

**Temperature Distribution $p_{\text{temp}}$.** The probability density of the temperature distribution $p_{\text{temp}}$ follows the Beta distribution as follows: $T = 2(1 + 0.2\alpha), \quad \alpha \sim \text{Beta}(\,\cdot\,; 1, 5)$.

The other hyperparameters are altered across the datasets and they are shown in Table 3. In addition, we also report the full list of hyperparameters used in training the baseline ensemble teacher networks in Table 4.

Table 3: The hyperparameter settings used to learn the DBN network.

| Dataset | CIFAR10 | CIFAR100 | TinyImageNet | ImageNet64 |
|---|---|---|---|---|
| #Params of Teacher | 1,860,986 | 7,460,708 | 21,798,504 | 21,798,504 |
| #Params of Score | 395,543 | 1,547,617 | 3,186,117 | 8,278,222 |
| Batch Size | 128 | 128 | 128 | 256 |
| Epochs | 800 | 800 | 250 | 250 |
| Epochs (distill) | 50 | 50 | 50 | 50 |
| Learning Rate | 0.00025 | 0.00025 | 0.0005 | 0.0005 |
| Learning Rate (distill) | 0.000025 | 0.000025 | 0.00005 | 0.00005 |
| Mixup $\alpha$ | 0.4 | 0.4 | 0.4 | 0.0 |
| $\beta_t$ | $0.0001, t \in [0,1]$ | $0.0001, t \in [0,1]$ | $0.001, t \in [0,1]$ | $0.005, t \in [0,1]$ |

Table 4: The hyperparameter settings used to learn the baseline ensemble models.

| Dataset | CIFAR10 | CIFAR100 | TinyImageNet | ImageNet64 |
|---|---|---|---|---|
| #Params | 1,860,986 | 7,460,708 | 21,798,504 | 21,798,504 |
| Batch Size | 256 | 128 | 128 | 256 |
| Epochs | 200 | 200 | 200 | 250 |
| Learning Rate | 0.1 | 0.1 | 0.1 | 0.1 |
| Cosine Decay Scheduling | Yes | Yes | Yes | Yes |
| Weight Decay | 0.001 | 0.0005 | 0.0005 | 0.0001 |
| Warmup Steps (Linear) | 0 | 5 | 5 | 5 |
| Initial Learning Rate | 0.1 | 0.001 | 0.001 | 0.001 |

# C GENERATION QUALITY OF DIFFUSION BRIDGE

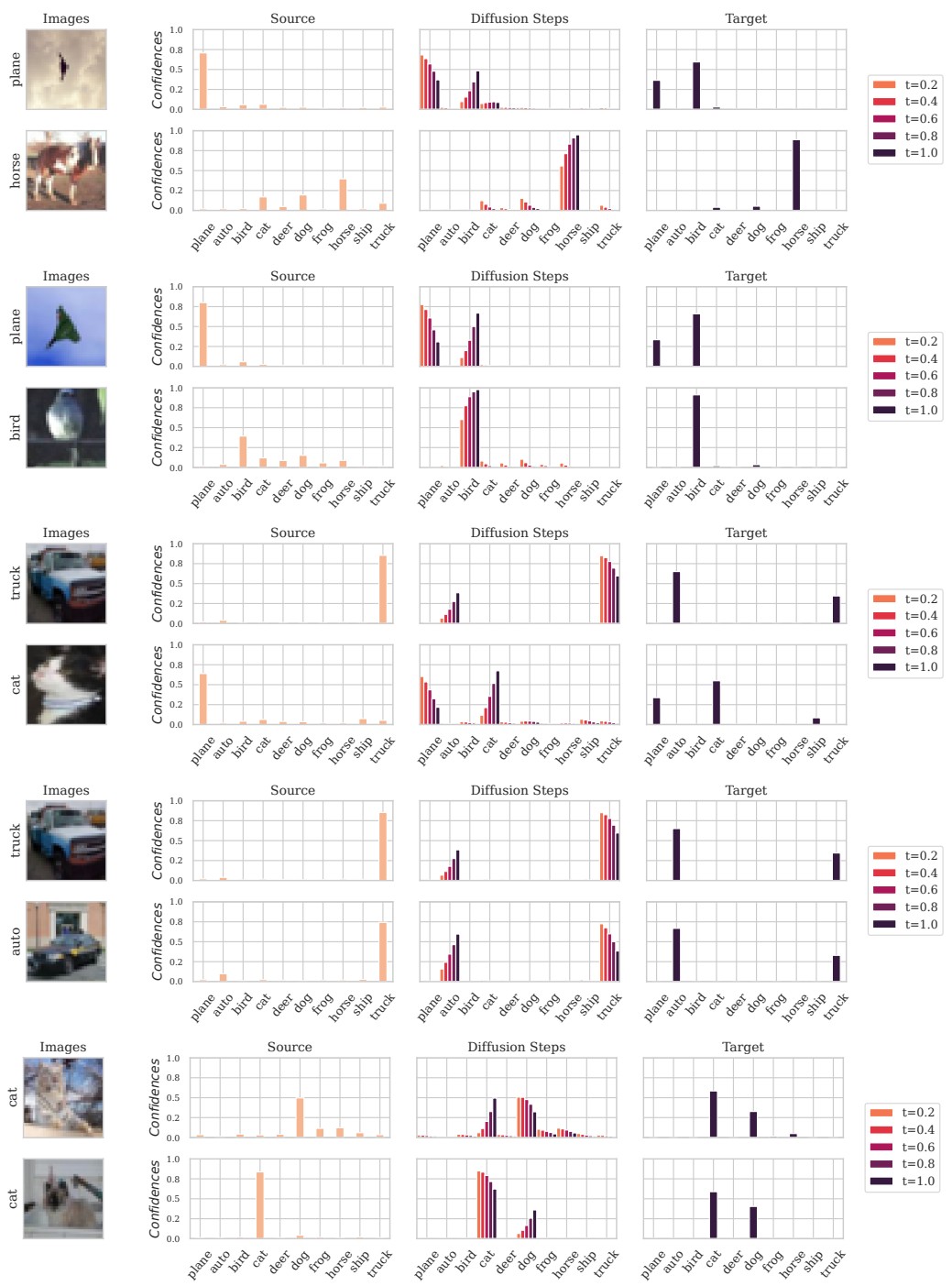

Figure 6: Confidences from the source model (first column), from the ensemble model (third column), and from the diffusion bridge (middle column) for the given images in the CIFAR-10 dataset. The middle column illustrates a transition of the confidence from the source model to the target during the diffusion process.

# D  ADDITIONAL EXPERIMENTS

## D.1  RESULTS ON IMAGENET

Because of the limit of the training budget, we conducted this on the downscaled datasets and obtained the following result. We also attached the results of BN in the baseline ImageNet and compared the DBN and BN with the baseline DEs independently. Then we observed that the performance of our DBN method surpasses DE-2 in terms of DEE and accuracy (ACC) and it is close to that of DE-3 with only one diffusion bridge, while BN achieves the same level of performance with two bridges.

Table 5: Performance on ImageNet dataset. cNLL stands for NLL calibrated with an optimal temperature.

| Model | FLOPs ($\downarrow$) | #Params ($\downarrow$) | ACC ($\uparrow$) | cNLL ($\downarrow$) | DEE ($\downarrow$) |
|---|---|---|---|---|---|
| ResNet (DE-1) | $\times$ 1.000 | $\times$ 1.000 | 66.47 | 1.3960 | 1.000 |
| +1 DBN (DE-3) | $\times$ 1.244 | $\times$ 1.380 | **69.79** | **1.2140** | **2.702** |
| DE-2 | $\times$ 2.000 | $\times$ 2.000 | 69.56 | 1.2542 | 2.000 |
| DE-3 | $\times$ 3.000 | $\times$ 3.000 | **70.82** | **1.1969** | **3.000** |

## D.2  OUT-OF-DISTRIBUTION PERFORMANCE

We evaluated our method on the widely used CIFAR-10-C (Hendrycks & Dietterich, 2019), which is the dataset of common corruptions on the CIFAR-10 dataset. and compared it to the existing BN method as in Table 1. Achieving superior performance in the CIFAR-10-C dataset implies that the model is robust with respect to the out-of-distribution distribution, tilted by a variety of distortions and corruptions in the in-distribution dataset (in this case, CIFAR-10).

Table 6: Test results on CIFAR-10-C. cNLL stands for NLL calibrated with an optimal temperature.

| Model | ACC ($\uparrow$) | NLL ($\downarrow$) | cNLL ($\downarrow$) |
|---|---|---|---|
| ResNet (DE-1) | 86.95 | 0.5158 | 0.4477 |
| +2 BN$_{medium}$ (DE-3) | 87.80 | **0.3740** | 0.3749 |
| +1 DBN (DE-3) | **88.91** | 0.3759 | **0.3682** |
| DE-2 | 88.85 | 0.3860 | 0.3693 |
| DE-3 | **89.47** | **0.3459** | **0.3410** |

# E  DEPTHWISE SEPARABLE CONVOLUTION

Depthwise separable convolution is a method to modify conventional convolutional layer, which include a convolution operation between every input channel and the convolution filters. Precisely, let the number of the input feature to have $H \times W$ spatial size with $C_{\text{in}}$ channels, and the convolutional layer with $h \times w$ receptive field with $C_{\text{out}}$ filters. For simplicity, we take full padding and do not allow strides,and biases. Then the number of parameters and the FLOPs of the convolutional layer is $h \times w \times C_{\text{in}} \times C_{\text{out}}$ and $H \times h \times W \times w \times C_{\text{in}} \times C_{\text{out}}$, respectively.

Compared to standard convolution, the depthwise separable convolution consists of two parts: *separable* convolution and *point-wise* convolution. First, *separable* convolution is the convolution with $h \times w$ receptive fields and $C_{\text{in}}$ filters, taking the number of groups same as the input feature ($C_{\text{in}}$). Then, each filter is convolved by the corresponding filters, followed by $H \times W \times C_{\text{in}}$ intermediate features. The number of parameters and the FLOPs of the separable convolution is $h \times w \times C_{\text{in}}$ and $H \times h \times W \times w \times C_{\text{in}}$, respectively. Then the second part of the depthwise separable convolution is the *point-wise* convolution, which is a $1 \times 1$ convolution with $C_{\text{in}}$ input and $C_{\text{out}}$ output filter sizes. The number of parameters and the FLOPs of the point-wise convolution is $C_{\text{in}} \times C_{\text{out}}$ and $H \times W \times C_{\text{in}} \times C_{\text{out}}$, respectively. We do not consider biases for evaluating the complexity of the convolution layers.

Table 7: The complexity measures of the standard convolution and depthwise separable convolution.

| Model | # Parameters | # FLOPs |
|---|---|---|
| Standard | $h \times w \times C_{\text{in}} \times C_{\text{out}}$ | $H \times h \times W \times w \times C_{\text{in}} \times C_{\text{out}}$ |
| Depthwise separable | $(h \times w + C_{\text{out}}) \times C_{\text{in}}$ | $(h \times w + C_{\text{out}}) \times H \times W \times C_{\text{in}}$ |
| D-S / Standard | $\dfrac{1}{C_{\text{out}}} + \dfrac{1}{hw}$ | $\dfrac{1}{C_{\text{out}}} + \dfrac{1}{hw}$ |

