# OpenReview forum: "Fast Ensembling with Diffusion Schrödinger Bridge"
_ICLR.cc/2024/Conference — ICLR 2024 poster_

### Official Review · Reviewer_3R6G · 2023-10-28

**Soundness:** 2 fair
**Presentation:** 2 fair
**Contribution:** 2 fair
**Rating:** 6
**Confidence:** 5

**Summary:**

The paper introduces a diffusion-based approach for enhancing Deep Ensemble (DE) performance. In particular, it suggests the direct transfer of samples from a chosen source model to the ensemble, utilizing the Image-to-Image Schrödinger bridge (I2SB). This proposed method seems to perform well on the conventional benchmark for ensembling tasks.

**Strengths:**

- The paper presents a compelling application of I2SB in the context of deep ensembles.

- The proposed method delivers impressive performance across various benchmarks, by achieving minimal performance degradation when compared to the ground-truth oracle, all while using significantly fewer FLOPs, making it an exciting development.

**Weaknesses:**

- Equation (4) contains an error: the forward and backward drifts should not be the same, and I2SB specifically provides a tractable solution only for the backward drift. Additionally, the definition of $f_t^\prime$ is missing. I suggest presenting the analytical backward drift directly, similar to what is provided for the target in (9), and ignore the forward diffusion.

- The variables "y" in (11) and (12) lack proper definitions. While their meanings can be inferred from the context, it would be better to provide clear definitions.

- The labels in Figure 2 are too small and may need to be enlarged for better readability.

- There appears to be a typo in (11). I think it should be "Softmax(Z_0)" instead of the current notation.

- Missing references in Sec 4.1

**Questions:**

- I believe I2SB remains applicable without the use of temperature annealing ($T$). Can the authors provide further clarification on the role of $T$ and explain how it contributes to differences in performance?

- It's unclear to me how multiple DBNs are combined in Sec 3.5. For instance, if there are 5 models (M=5) and each DBN is trained with 3 ensembles (as suggested in Sec 4), what is the appropriate value for $L$ and what're the ensembles for each DBN?

- I2SB has demonstrated decent performance with NFE=1. Can the authors present an ablation study or provide justification for why additional distillation is necessary for achieving optimal performance?

- The font size in Alg 1 seems to be smaller than the one in the main paper. Is this intended?

I am open to reevaluating the score, with the condition that the authors thoroughly address my questions and make improvements to the presentation, particularly addressing the weaknesses I've highlighted.

---

> ### Author Response · Authors · 2023-11-17
> **Response to Reviwer 3R6G (1)**
>
> __Clarity of the diffusion bridge background__
> >Equation (4) contains an error: the forward and backward drifts should not be the same, and I2SB specifically provides a tractable solution only for the backward drift. Additionally, the definition of $f’_t$ is missing. I suggest presenting the analytical backward drift directly, similar to what is provided for the target in (9), and ignore the forward diffusion.
>
> Thank you for your suggestions! As you mentioned, the forward and backward drifts are generally not the same in the Schrödinger equation, and I2SB provides a tractable solution only for the backward drift, and only the backward drift is required for evaluation. In Section 2.2 of the revised manuscript, we compensated the DSB background by describing the full procedures of the Schrödinger bridge as an optimal transport problem w.r.t. the forward and backward SDEs with two drifts, and how the backward drift is simulated by the score network in Section 3.2. We also added the explanation on the Fokker-Planck equation for completeness, since the tractable solution for the backward drift is obtained by solving the time-reversal Fokker-Planck equation, leading $\Phi’$ to be tractable, which was generally intractable in existing SB problems before simplifying into the conditional distribution.
> * * *
> __Clarification of the writing__
> >The variables "y" in (11) and (12) lack proper definitions. While their meanings can be inferred from the context, it would be better to provide clear definitions.
> The labels in Figure 2 are too small and may need to be enlarged for better readability.
> The font size in Alg 1 seems to be smaller than the one in the main paper. Is this intended?
> There appears to be a typo in (11). I think it should be "Softmax(Z_0)" instead of the current notation.
> Missing references in Sec 4.1
>
> Thank you for the careful reading and feedback on our paper. We sincerely apologize for the confusion in our paper. We clarified the manuscript based on the feedback by the following:
> 1. We provided the definition y in (11) and (12), which is colored blue in the revised manuscript. (In the revised manuscript, these equations are now (12) and (13).)
> 2. For the labels in Figure 2, we will enlarge the labels in the Figure in the camera-ready version of the paper.
> 3. It is intended that Algorithm 1 uses a smaller font size than the main paper, for fitting the page limit of the conference.
> 4. The Softmax(Z_1) is a typo and is fixed to Softmax(Z_0). In addition, since the subscript $z_i$ is duplicated in the time domain of the intermediate logits and the ensemble outputs, we super-scripted the ensemble output logits into $z^{(i)}$.
> 5. The missing references in Section 4.1 are removed since the references had been linked to non-existing tables in our LaTeX file.
> * * *
> __Temperature annealing__
> > I believe I2SB remains applicable without the use of temperature annealing ($T$).
>
> > Can the authors provide further clarification on the role of $T$ and explain how it contributes to differences in performance?
>
> Thank you for the insightful question and remarks on temperature annealing! The random annealing is necessary to prevent the model from falling into a trivial solution. We detailed the role of the temperature annealing $T$ in **Common Response 3**.
> * * *
> __Combining multiple DBNs__
> > It's unclear to me how multiple DBNs are combined in Sec 3.5. For instance, if there are 5 models (M=5) and each DBN is trained with 3 ensembles (as suggested in Sec 4), what is the appropriate value for $L$ and what're the ensembles for each DBN?
>
> For more than 3 ensembles, suppose that we have 5 ensemble members $\theta_1,\ldots,\theta_5$. For the first DBN, we choose $\theta_1$ as a source model and distill 3 ensembles: $\theta_1,\theta_2,\theta_3$. The second DBN also has to use $\theta_1$ as a source model and distill other 3 ensembles including the source model: $\theta_1,\theta_4,\theta_5$. When we do inference, we first run the source model $\theta_1$ and get the source logit. Two different DBNs produce different outputs from the source logit. The final output is obtained by averaging the two outputs, as denoted in Equation (13). We simply revised the paper to elaborate the inference with multiple DBNs, colored blue in Section 3.5 in the paper. Due to the page limit, we are planning to add the examples in the final version.
>
> For the conventional ensemble distillation methods, it is well known that the number of ensembles that can be distilled via a single student model is saturated to a certain level. Figure 5 in Section 4.3 shows the similar property in the DBN method and the number of ensembles that a single DBN can distill was saturated to 3, so we fixed the number of ensemble members that DBN distills to three. When we need to distill more than 3 ensembles, adding DBN is more efficient than increasing the size of the DBN network because we can expect additional ensemble effects by averaging outputs of $N$ number of DBNs.

---

> ### Author Response · Authors · 2023-11-17
> **Response to Reviewer 3R6G (2)**
>
> __On distillation of I2SB__
> > I2SB has demonstrated decent performance with NFE=1. Can the authors present an ablation study or provide justification for why additional distillation is necessary for achieving optimal performance?
>
> We apologize for any confusion caused by our paper. The I2SB paper showcases strong performance with NFE equal to or greater than ten, with a noticeable drop in performance for smaller numbers of function evaluations, such as NFE=2 (refer to Figure 8 in [1] for more details). Based on the findings in [1], we assumed that near-optimal performance for I2SB is achieved at around NFE=10, and consequently, we set the initial number of timesteps to 5 for both training and inference (For additional NFE values like 10 or 15, please refer to **Table 1**). The primary reason for employing diffusion distillation is to reduce the inference cost while maintaining performance at an appropriate level.
>
> **Table 1**. DBN on CIFAR-10 with different number of inference steps.
> |                  | ACC   | NLL    |
> |------------------|-------|--------|
> | DE-1             | 91.30 | 0.3382 |
> | +1 DBN (step 5)  | 92.97 | 0.2435 |
> | +1 DBN (step 10) | 93.01 | 0.2441 |
> | +1 DBN (step 15) | 92.95 | 0.2442 |
>
> [1] G. Liu et al., “I2SB: Image-to-Image Schrödinger Bridge”, ICML 2023, URL: https://arxiv.org/abs/2302.05872

---

> > ### Comment · Reviewer_3R6G · 2023-11-20
> > **Reviewer Response**
> >
> > I first thank the author’s efforts in their responses, which I found helpful. There’re a few issues that need to be addressed:
> >
> > - In Eq 4, I suggest to use $z$ rather than the capitalized letter $Z$, which is typically denoted for random variable. Eq 4 are PDE that defines evolution of function, and there is no inherent randomness involved. This adjustment aids in clarity, as it helps avoid confusion between whether $Z$ refers to the solution to (3) or (5), which is inaccurate in either case.
> > - It is incorrect to say “(4) and its time-reversal directly follows the Fokker-Planck
> > equation of the SDE in (3) are as follows”. (4) indeed represents the FPE of a family of SDEs, of which (5) is one example, but not (3).
> > - Can the author add “+1 DBN (step 1)” to Table 1? Given that I2SB is constructed in a latent space with much fewer dimensions compared to Fig 8 in [1], I still suspect NFE=1 should do just fine, especially since DE-1 already performs sufficiently well (meaning the two boundary distributions are quite close to each other).

---

> > > ### Author Response · Authors · 2023-11-21
> > > **Re: Response to Reviewer 3R6G**
> > >
> > > __Clarity of notation__
> > >
> > > We again appreciate giving thoughtful reading and feedback on our paper. We will again clarify our paper, according to the following parts. I hope the following refinement clarifies the writing and help the readers for better understanding.
> > >
> > > > In Eq 4, I suggest to use $z$ rather than the capitalized letter $\boldsymbol{Z}$, which is typically denoted for random variable. Eq 4 are PDE that defines evolution of function, and there is no inherent randomness involved. This adjustment aids in clarity, as it helps avoid confusion between whether $\boldsymbol{Z}$ refers to the solution to (3) or (5), which is inaccurate in either case.
> > >
> > > We modified the $z$ in Equation (4) to $\boldsymbol{z}$, as each $\boldsymbol{z}_t$ denotes the particle at time $t$.
> > >
> > > > It is incorrect to say “(4) and its time-reversal directly follows the Fokker-Planck equation of the SDE in (3) are as follows”. (4) indeed represents the FPE of a family of SDEs, of which (5) is one example, but not (3).
> > >
> > > Thank you for pointing out the reference and we apologize for this error. We corrected (3) to (5) in the revised version of the paper.
> > >
> > > * * *
> > >
> > > __Ablation on inference step 1__
> > >
> > > > Can the author add “+1 DBN (step 1)” to Table 1? Given that I2SB is constructed in a latent space with much fewer dimensions compared to Fig 8 in [1], I still suspect NFE=1 should do just fine, especially since DE-1 already performs sufficiently well (meaning the two boundary distributions are quite close to each other).
> > >
> > > Thank you for your response. We have added the diffusion bridge trained with a one-step model (+1 DBN (step 1)) as demonstrated in __Table 2__. According to __Table 2__, the one-step model exhibits comparable performance in terms of accuracy, but slightly lower performance in terms of likelihood.
> > >
> > > __Table 2__. DBN on CIFAR-10 with different number of inference steps, including [step 1] case.
> > > |                  | ACC   | NLL    |
> > > |------------------|-------|--------|
> > > | DE-1             | 91.30 | 0.3382 |
> > > | +1 DBN (step 1)  | 92.97 | 0.2469 |
> > > | +1 DBN (step 5)  | 92.97 | __0.2435__ |
> > > | +1 DBN (step 10) | __93.01__ | 0.2441 |
> > > | +1 DBN (step 15) | 92.95 | 0.2442 |

---

> > > > ### Comment · Reviewer_3R6G · 2023-11-21
> > > > **Reviewer Response**
> > > >
> > > > Thanks. I have increased the rating to reflect the discussions at the current stage.

---

> ### Author Response · Authors · 2023-11-21
> **Re: Re: Response to Reviewer 3R6G**
>
> We greatly appreciate the active discussion of the reviewer and re-evaluating our paper!

---

### Official Review · Reviewer_jMzV · 2023-10-31

**Soundness:** 3 good
**Presentation:** 3 good
**Contribution:** 3 good
**Rating:** 8
**Confidence:** 3

**Summary:**

In this manuscript, the authors endeavor to address the limitations of existing bridge networks by introducing a novel "diffusion bridge networks" framework, inspired by the celebrated Schr\"odinger Bridge problem. Specifically, the authors commence by elucidating the shortcomings inherent in constructing bridges within the feature space represented by bridge network, and subsequently propose their diffusion bridge as a solution to generalize from a single network output to an ensemble output. In order to empirically substantiate the efficacy of their method, a variety of experiments are executed across the CIFAR-10, CIFAR-100, and TinyImageNet datasets, utilizing ResNet-$32\times2$, ResNet-$32\times4$, and ResNet-$34$ as backbone architectures. Overall, the paper offers an intuitive approach to the problem at hand, and I will delineate its strengths and weaknesses in the subsequent sections.

**Strengths:**

1. **Problem Reformulation** In the domain of problem formulation, the authors ingeniously recast the challenge of deep ensembles as an optimal transport problem, subsequently addressing it through a dynamical optimal transport methodology. This novel perspective not only refines the research question but also paves the way for utilizing advanced mathematical tools to provide robust solutions.

3. **Ample Experiments** With regard to empirical validation, the authors carry out an extensive array of experiments on multiple datasets and furnish an in-depth analysis of the experimental outcomes. This comprehensive experimental setup serves to fortify the credibility of their proposed method and provides valuable insights into its performance characteristics.

**Weaknesses:**

1. **Other Transport Models** To the best of my understanding, related works such as "Rectified Flow" [1] and "Flow Matching" [2] are capable of accomplishing similar functionalities. Additionally, when compared to the Schrödinger bridge approach, these models operate through a deterministic process, which could potentially benefit from lower variance properties. The omission of such deterministic flow models from the discussion may limit the generalizability and applicability of the method proposed in the current manuscript.

0. **Inference Algorithm** In the manuscript, it appears that the authors have not provided details regarding the inference algorithm for the proposed DBN during the model inference stage. This omission leaves a critical gap in the paper, as understanding the inference mechanism is essential for a comprehensive evaluation of the proposed method.

2. **Annealing of Temperature** In Section 3.2, the authors discuss the concept of temperature distribution, yet the experimental section lacks elaboration on how the temperature distribution is selected and what principles govern temperature annealing. This absence of information creates a gap in the methodological clarity and poses questions regarding the thoroughness of the experimental design. Furthermore, from a scholarly perspective, treating the distribution as a Gumbel-Softmax distribution [3] could raise additional questions. Specifically, one might inquire whether the training variance of the Diffusion Bridge Networks (DBN) is influenced by the temperature parameter. Addressing such intricate relationships between the temperature and training variance would enhance the paper's academic rigor and contextual relevance.

3. **LaTeX Compile** In Section 4.1, line 3, the manuscript contains two instances of "??", which clearly indicate placeholders or unresolved references. The authors should rectify this issue to enhance the document's professionalism and completeness prior to submitting the finalized manuscript. The presence of such markers detracts from the paper's overall quality and could create potential ambiguities for the reader.
----
References:
1. Liu X et al. Flow Straight and Fast: Learning to Generate and Transfer Data with Rectified Flow, ICLR 2023
2. Lipman Y et al. Flow Matching for Generative Modeling, ICLR 2023
3. Jang E,  et al. Categorical Reparameterization with Gumbel-Softmax, ICLR 2017

**Questions:**

My questions are listed in weakness, please refer to weakness.

----
Post Rebuttal Comments:
Thank you for the comprehensive response to my queries and for conducting additional experiments. Having thoroughly reviewed your revised manuscript and the expanded experimental evidence, I am inclined to revise my evaluation. Accordingly, I am considering raising my score from a 6 to an 8, reflecting the improvements and clarifications you have made.

---

> ### Author Response · Authors · 2023-11-17
> **Response to Reviewer jMzV**
>
> __Comparison to the generative flow approaches__
> >To the best of my understanding, related works such as "Rectified Flow" [1] and "Flow Matching" [2] are capable of accomplishing similar functionalities. Additionally, when compared to the Schrödinger bridge approach, these models operate through a deterministic process, which could potentially benefit from lower variance properties. The omission of such deterministic flow models from the discussion may limit the generalizability and applicability of the method proposed in the current manuscript.
>
> We sincerely appreciate you for insightful questions! As you have issued, the deterministic flow models that simulate the ODE that bridges different distributions have similar functionalities and benefit from variance reduction properties. Nevertheless, adding some stochasticity in the learning process leads us to avoid the probability flow to fall into the trivial solution. This is also explained as we further describe the necessity of the temperature annealing for robustifying the source datasets, which is described in Common Response 3 in detail. The result that compares our approach to deterministic flow models using rectified flow is provided in **Common Response 2**.
> * * *
> __The inference algorithm__
> >In the manuscript, it appears that the authors have not provided details regarding the inference algorithm for the proposed DBN during the model inference stage. This omission leaves a critical gap in the paper, as understanding the inference mechanism is essential for a comprehensive evaluation of the proposed method.
>
> We apologize for giving confusion on the inference algorithm. We first appended how the feature extractor $g_{\psi_1}$ computes the feature $\boldsymbol{h}_1$ by forward evaluation, and the inference algorithm is a single step of reverse-time evaluation of the score network via the distilled score network from time $1\rightarrow 0$, as described in Equation (12). We also added the detailed architecture of the score network utilized for inference in the Figure 5 of Appendix A.2, for further understanding.
> * * *
> __Temperature annealing__
> > In Section 3.2, the authors discuss the concept of temperature distribution, yet the experimental section lacks elaboration on how the temperature distribution is selected and what principles govern temperature annealing. This absence of information creates a gap in the methodological clarity and poses questions regarding the thoroughness of the experimental design.
>
> Thank you for giving insightful questions and suggestions about the temperature annealing, and we apologize for causing confusion on the concept of temperature distribution. For our choice of the temperature distribution, we chose the Beta distribution $T = 2 ( 1+0.2\alpha), \alpha\sim \mathrm{Beta}(\cdot; 1, 5)$ for simple and efficient temperature annealing. We added these details on hyperparameters including the temperature annealing in Appendix B. For further details on temperature annealing, please refer to **Common Response 3**.
>
> > Furthermore, from a scholarly perspective, treating the distribution as a Gumbel-Softmax distribution [3] could raise additional questions. Specifically, one might inquire whether the training variance of the Diffusion Bridge Networks (DBN) is influenced by the temperature parameter. Addressing such intricate relationships between the temperature and training variance would enhance the paper's academic rigor and contextual relevance.
>
> We appreciate the suggestion that can support the academic rigor and contextual relevance by improving theoretical backgrounds on our proposed methods. We will investigate these theoretical aspects and update this in the final version of our paper.
> * * *
> __Clarity of the paper__
> > In Section 4.1, line 3, the manuscript contains two instances of "??", which clearly indicate placeholders or unresolved references. The authors should rectify this issue to enhance the document's professionalism and completeness prior to submitting the finalized manuscript. The presence of such markers detracts from the paper's overall quality and could create potential ambiguities for the reader.
>
> Thank you for pointing out the errors. We sincerely apologize for the completeness of our paper, by mis-referencing and errors in our manuscript. We intensively looked over the manuscript and fixed the typo, grammar errors, and the duplicating notations. We also modified the two reference instances in the manuscript.

---

> ### Author Response · Authors · 2023-11-21
> **Response to Reviewer jMzV**
>
> We greatly appreciate to reviewing our manuscript and re-evaluating the score of our paper! We will further resolve the suggested points in the final version of the paper.

---

### Official Review · Reviewer_Qbom · 2023-11-01

**Soundness:** 2 fair
**Presentation:** 2 fair
**Contribution:** 2 fair
**Rating:** 6
**Confidence:** 3

**Summary:**

This article presents a framework to approximate Deep Ensembles (DE) called Diffusion Bridge Network (DBN). Following the theory of the Schrödinger bridge, a conditional diffusion bridge is learned between single models' logits and the target ensemble's logits. The score network learned is distilled during the training process and multiple score networks can be ensembled when the number of models increases.

[Edit: as indicated in my response, I am increasing my rating from 5 to 6 considering the strong improvements in writing for both the theory and the implementation, despite limited results on TinyImageNet after correction of the authors mistake]

**Strengths:**

The main idea of the article is well-motivated, improving on Bridge Networks with the strong Schrödinger bridge to learn the ensemble distribution.

The results obtained show that the conditional diffusion bridge can be learned, leading to efficient deep ensemble estimation, with a strong Deep Ensemble Equivalent score. However with limited insight on the networks used and their training, it is hard to conclude and compare with Bridge Networks.

**Weaknesses:**

**Writing** This article's writing is very hard to follow. There are many spelling or grammar mistakes (see Questions). Ideas are presented poorly or too quickly, leaving the reader confused. The theory, which originates from Liu et al. 2023 is presented very quickly and unclearly, with some mistakes. ($f'$?, no definitions of $W_t$, $\bar W_t$, $\beta_t$, no dimensions, Schrödinger systems are not presented, it's never clarified that $f=0$, although $f$ was never even defined. It's not clear how equation (8) is obtained.

**Training** Many important points are missing, in particular, the architecture of the score network is never given anywhere in the article. Contrary to what the authors claim, the appendix does not give any hyperparameter settings: the training details of both the ensemble networks and the DBN are left unclear (Optimizer, scheduler, number of epochs/steps, distillation schedule...) This is not a detail, in particular for the DBN. If its training is particularly expensive/slow, it is an important drawback despite the speedup it provides during inference. Are Bridge Networks also using distillation, and is their training comparable to DBN? Without this information, it is hard to compare the results obtained.

**Number of ensembles** I find unclear the number of ensembles that a score network is trained on. Section 4 indicates that a score network is trained on 3 ensembles, with no clarification on why this number is chosen (presumably Section 4.3). Why also not simply increase the size of the DBN network? For more than 3 ensembles, 2 or more DBNs are trained. In this case, are the DBN trained on all of the ensembles at the same time? Are they limited to a given subset of 3 ensembles, as indicated previously?
This becomes even more confusing in Section 4.2, where the number of ensembles increases to 9, and the number of DBNs is left unclear ("two or more"), as well as for BNs.

Despite promising results, this article needs a serious rewrite before I can accept it for now.

**Questions:**

* Is the DSB not able to learn if there is no temperature annealing?

* Why choose in particular ResNet-32x2 and x4 rather than a standard ResNet-32?

* Do the authors have an idea why DBN shows poor EC scores compared to the other metrics?

**Various mistakes or remarks:**
* Abstract: "theory of [the] Schrödinger bridge"
* Introduction: "incurrs an extra"
* 2.2 "has rarely been demonstrated its practicality" "demonstrated for real-world image dataset" "Gaussain" "by a certain PDEs"
* 3.2 "Based on the formulation" ??
* 3.3 "super performance". No use for the (NFE) abbreviation if it's never used after.
* 3.4 $\Phi$ or $\Phi'$?
* 4. "because Ryabinin et al. (2021) refined poor convergence of" "more profound formulations"
* 4.1 "results on CIFAR-10 is shown in ?? of ??"
* Conclusion "Additionally,  [...] of 3 DE models", nominal sentence.

---

> ### Author Response · Authors · 2023-11-17
> **Response to Reviewer Qbom (1)**
>
> __Clarification of the writing - Diffusion Schrodinger bridge__
>
> > This article's writing is very hard to follow. There are many spelling or grammar mistakes (see Questions). Ideas are presented poorly or too quickly, leaving the reader confused. The theory, which originates from Liu et al. 2023 is presented very quickly and unclearly, with some mistakes. ( $f ′$ ?, no definitions of  $W_t$ ,  $\tilde{W}_t$ ,  $β_t$ , no dimensions, Schrödinger systems are not presented, it's never clarified that  $f = 0$, although  $f$  was never even defined. It's not clear how equation (8) is obtained.
>
> Thank you for the thoughtful reading and concerns on our paper. We apologize for the clarity of this paper and made up the Schrodinger bridge part (Section 2.2) of our paper. Briefly, we complemented the setting of the Schrodinger systems and denoted the variables that are not defined. And we added some supporting statements such as how I2SB solves the restoration problem in the background part. And we also clarified the equation (8) (The main SDE from $Z_1$ to $Z_0$) in the paper.
>
> For further details, please refer to the revised version (the refined parts written in blue) of Sections 2.2 and 3.2 in our paper. We also clarified the list of the revised parts in **Common Response 4**.

---

> ### Author Response · Authors · 2023-11-17
> **Response to Reviewer Qbom (2)**
>
> __Detailed architectures and hyperparameter settings__
> > Many important points are missing, in particular, the architecture of the score network is never given anywhere in the article. Contrary to what the authors claim, the appendix does not give any hyperparameter settings: the training details of both the ensemble networks and the DBN are left unclear (Optimizer, scheduler, number of epochs/steps, distillation schedule...) This is not a detail, in particular for the DBN. If its training is particularly expensive/slow, it is an important drawback despite the speedup it provides during inference. Are Bridge Networks also using distillation, and is their training comparable to DBN? Without this information, it is hard to compare the results obtained.
>
> We apologize for the absence of details regarding the architectures and settings of the DBN model and thanks for pointing them out. The training details of the DBN are outlined below.
>
> 1. **Architecture of the Score Network**
>
> Our score network initially embeds three inputs: $Z_t$ (the input logit at time $t$), $h_1$ (the condition feature), and time $t$. Subsequently, the three embedded features of the inputs pass through the score network, as depicted in Figure 5, consisting of multiple InvertedResidual blocks. A detailed architecture description of the network is provided in Appendix A.2.
>
> 2. **Hyperparameter Settings and Training Details of the Ensemble Networks and DBN**
>
> In conjunction with the score network architecture, we have included the hyperparameter settings of our model in Appendix B. The detailed hyperparameters, listed in Table 2, encompass the number of parameters of the classifier and score networks, batch size, epochs, and learning rates for the main training and distillation phases, mixup coefficients used for data augmentation, and the diffusion drift coefficient for generating the diffusion bridge.
>
> Additionally, we elaborate on the omitted hyperparameter settings of the ensemble networks, referred to in Table 4 of Appendix B, which were simply assumed to use the same architecture as the BN paper.
>
> 3. **Comparison of BN to DBN**
>
> To be more precise, BN distills the output of a model located along the low-loss subspace between teacher networks. As we described in Section 2.1, the bridge network [1] hinges on the features of the mode connectivity among the low-loss subspace. BN first learns the pre-trained low-loss surfaces between two pre-trained ensemble members by learning the “anchor” parameter that satisfies on the Bezier curve, i.e., finding the curve with low loss on the loss surface. BN assumed and empirically found that the midpoint of the Bezier curve tends to achieve the lowest loss and used this midpoint (parameterized by $\theta_{i,j}(0.5)$) as their teacher. Therefore, they train a lightweight neural network that distills the midpoint.
>
> During the training process, the size of the teacher network is significantly larger than that of the Bridge in both BN and DBN. Besides, DBN uses small diffusion steps, the majority of the training cost arises from calculating the outputs of the teacher networks for distillation, so **the cost of learning the Bridge of both BN and DBN is relatively negligible compared to the teachers**.
>
> To elaborate further, assuming N pre-trained weights for the Teacher model (cost=$1$), the training process for BN and DBN to distill 3 teacher models (DE-3) is as follows. First, BN selects 2 pairs of teachers from three teacher weights and learns $2$ low-loss curves connecting the teacher pairs using the teacher architecture (cost=$2\times1$). Then, it additionally trains $2$ BNs to predict the output of the model in the middle ($\theta_{i,j}(0.5)$) of each low-loss curve (cost=$2\times2.xx$). Similarly, for DBN, it selects 1 source teacher and trains $1$ score network (smaller than BN) to learn the ensemble outputs of $3$ teachers with 5 diffusion steps (cost=$1\times3.xx$). Following this, an additional training is conducted where the 5 steps are distilled into one step (cost=$1\times3.xx$). In the final version of the paper, we will provide a clearer explanation of the training cost of DBN relative to BN.
>
> [1] E. Yun et al., “Traversing between modes in function space for fast ensembling”, ICML 2023
>
> [2] T. Salimans et al., “Progressive distillation for fast sampling of diffusion models”, ICLR 2022

---

> ### Author Response · Authors · 2023-11-17
> **Response to Reviewer Qbom (3)**
>
> __Training details of the DBN models__
> > I find unclear the number of ensembles that a score network is trained on. Section 4 indicates that a score network is trained on 3 ensembles, with no clarification on why this number is chosen (presumably Section 4.3). Why also not simply increase the size of the DBN network? For more than 3 ensembles, 2 or more DBNs are trained. In this case, are the DBN trained on all of the ensembles at the same time? Are they limited to a given subset of 3 ensembles, as indicated previously? This becomes even more confusing in Section 4.2, where the number of ensembles increases to 9, and the number of DBNs is left unclear ("two or more"), as well as for BNs.
>
> For the conventional ensemble distillation methods, it is well known that the number of ensembles that can be distilled via a single student model is saturated to a certain level. Figure 5 in Section 4.3 shows the similar property in the DBN method and the number of ensembles that a single DBN can distill was saturated to 3, so we fixed the number of ensemble members that DBN distills to three. When we need to distill more than 3 ensembles, adding DBN is more efficient than increasing the size of the DBN network because we can expect additional ensemble effects by averaging outputs of $N$ number of DBNs.
>
> For more than 3 ensembles, suppose that we have 5 ensemble members $\theta_1,\ldots,\theta_5$. For the first DBN, we choose $\theta_1$ as a source model and distill 3 ensembles: $\theta_1,\theta_2,\theta_3$. The second DBN also has to use $\theta_1$ as a source model and distill other 3 ensembles including the source model: $\theta_1,\theta_4,\theta_5$. When we do inference, we first run the source model $\theta_1$ and get the source logit. Two different DBNs produce different outputs from the source logit. The final output is obtained by averaging the two outputs, as denoted in Equation (13). We simply revised the paper to elaborate the inference with multiple DBNs, colored blue in Section 3.5 in the paper. Due to the page limit, we are planning to add the examples in the final version.
>
> > Is the DSB not able to learn if there is no temperature annealing?
>
> Thank you for the thoughtful question! Since the issue of temperature annealing in the source logits is a common issue over reviewers, we provided the interpretation and additional explanations on why we made use of temperature annealing and why this is necessary in the **Common Response 3**.
>
> > Why choose in particular ResNet-32x2 and x4 rather than a standard ResNet-32?
>
> In order to fairly compare the performance between DBN and the existing BN method, we followed the architecture that the BN paper used. For CIFAR-10 and CIFAR-100 datasets, the BN paper used the ResNet-32x2 and ResNet-32x4 architecture.
>
> > Do the authors have an idea why DBN shows poor EC scores compared to the other metrics?
>
> We explained about the issues on the expected calibration error (ECE) metric in **Common Response 4**.
> * * *
> __Clarification of writing - Typos and grammar mistakes__
>
> > Fixing typos and grammar mistakes:
>
> > “theory of [the] Schrödinger bridge” (abstract)
>
> > "incurrs an extra" (introduction)
>
> > "has rarely been demonstrated its practicality" "demonstrated for real-world image dataset" "Gaussain" "by a certain PDEs" (section 2.2)
>
> > "Based on the formulation" ?? (section 3.2)
>
> > "super performance". No use for the (NFE) abbreviation if it's never used after. (section 3.3)
>
> > $\Phi$ or $\Phi’$? (section 3.4)
>
> > "because Ryabinin et al. (2021) refined poor convergence of" "more profound formulations" (section 4)
>
> > "results on CIFAR-10 is shown in ?? of ??" (section 4.1)
>
> > "Additionally, [...] of 3 DE models", nominal sentence. (conclusion)
>
> We sincerely apologize for the errors we have made in the manuscript. All these points are fixed in the revised manuscript.

---

> ### Author Response · Authors · 2023-11-23
> **Remainder for further discussion**
>
> Dear reviewer Qbom,
>
> We sincerely hope that our response successfully addressed your concerns and issues considered.
> We found that the thoughtful comments and feedbacks had been an invaluable help in improving our manuscript.
>
> As the deadline of the author-reviewer discussion period is approaching, we would be grateful to have further discussions until the discussion period ends.
>
> Best regards, \
> Authors

---

### Official Review · Reviewer_bFwN · 2023-11-02

**Soundness:** 3 good
**Presentation:** 3 good
**Contribution:** 3 good
**Rating:** 8
**Confidence:** 4

**Summary:**

This work proposes to lower the inference costs associated with running a full ensemble of neural networks.
One problem with other fast ensembling approaches is that they require mode connectivity between the ensemble members, but finding such a low-loss path or subspace through the weight space is difficult. other methods either assume the location of these parameters directly, or else learning-based approaches learn a subspace between just two modes (ensemble members). This method uses a diffusion model to transform logits from a single ensemble member, into a sample from the logit distribution of the ensemble itself.

**Strengths:**

I think this is a strong approach, it alleviates inference costs in a straightforward way that makes use of novel techniques.

* Interesting approach to speed up ensemble inference
* Frames ensemble knowledge distillation as a diffusion problem
* Linear scaling of bridges in the worst case, where bridges are smaller than ensemble members.
* Good results on accuracy across benchmarks. DEE is very good.

**Weaknesses:**

* DBN models are notably less calibrated  than ensembles or bridge networks across tasks.
* No evaluation on in / out of domain uncertainty / calibration. IMO this is a huge reason why ensembling is done in the first place.
* Evaluation is with relatively small models

Comments but not weaknesses:
* Section 4.1 "?? of ??" still in text.
* I think parameter-count/memory savings is useful here and could be reported.


---------------------------------------------------------------------------------------------
Post Rebuttal Comments
---------------------------------------------------------------------------------------------
After reading the authors responses and the other reviews I will not be updating my score.
Detailed comment below.

I think the authors addressed most of my concerns in their comments to my and other reviews.

* W2 and W3 have been mostly addressed with additional experiments
* Q1 was answered empirically, though I would hope any accompanying text came with intuition about why/if loss should decrease monotonically with increased t
* Q2 was brought up by most other reviewers, and the common response is fine, if not somewhat "hand-wavey". Nonetheless, its not a glaring concern and improvement could be a component of future work.
* Q3/4/4a/5 were answered

**Questions:**

1. Given a path from model z_1 to the ensemble model z, can anything be said about the intermediate logit distributions in terms of their loss?
2. I'm not clear how useful the source distribution construction is when its just annealing T. This preserves relative ordering over classes, which is not adding much additional information about the "distribution" of z_1. Are we in some sense integrating over paths from the set of temperature-augmented z_1 -> z?  Is that a non-trivial path?
3. In algorithm 1: is a single temperature value drawn for each member, for all data points in the minibatch?
4. Does performance increase with multiple diffusion steps?
4a. What's the "cost" of each diffusion step in term of % of DE inference speed. Can DBN admit more than diffusion step at inference and still be faster than DE?
5. Can the authors comment on the calibration error increase with this method?

---

> ### Author Response · Authors · 2023-11-17
> **Response to Reviewer bFwN (1)**
>
> We appreciate your sincere consideration and insights, and positive comments on our paper! We address the raised issues in detail as outlined below.
>
> __Uncertainty evaluation__
>
> > No evaluation on in / out of domain uncertainty / calibration. IMO this is a huge reason why ensembling is done in the first place.
>
> We evaluated our method on the widely used CIFAR-10-C [1], which is the dataset of common corruptions on the CIFAR-10 dataset. and compared it to the existing BN method as in __Table 1__.  Achieving superior performance in the CIFAR-10-C dataset implies that the model is robust with respect to the out-of-distribution distribution, tilted by a variety of distortions and corruptions in the in-distribution dataset (in this case, CIFAR-10). We will update the full OOD results in the final version.
>
> Table 1. Test results on CIFAR-10-C. cNLL stands for NLL calibrated with an optimal temperature.
>
> |        | ACC   | NLL    | cNLL   |
> |--------|-------|--------|--------|
> | DE-1   | 86.95 | 0.5158 | 0.4477 |
> | +2 BN  | 87.80 | 0.3740 | 0.3749 |
> | +2 DBN | 88.91 | 0.3759 | 0.3682 |
> | DE-2   | 88.85 | 0.3860 | 0.3693 |
> | DE-3   | 89.47 | 0.3459 | 0.3410 |
>
> [1] D. Hendrycks, “Benchmarking Neural Network Robustness to Common Corruptions and Perturbations”, ICLR 2019
>
> * * *
>
> __Extended experiments__
>
> > Evaluation is with relatively small models.
>
> Due to the lack of our computational budget, there was some limit on scaling up the neural network model. Instead, we tried our model on more complex and large-scale datasets, ImageNet64. We included the extended experimental results in __Common Response 1__.
>
> > I think parameter-count/memory savings is useful here and could be reported.
>
> In our paper, the ratio of saved memory compared to the deep ensemble models is introduced in the #Params column of Table 1 of the paper. The actual number of parameters (and thus the number of saved parameters) is included in the Appendix B.
>
> > Given a path from model z_1 to the ensemble model z, can anything be said about the intermediate logit distributions in terms of their loss?
>
> We provided a brief illustration of the intermediate logits from the source to the target in the probability space, as shown in Figure 5 of Appendix B. Alongside the qualitative analysis, __Table 2__ below showcases the evolution of accuracy and NLL for a single DBN in the CIFAR-10 dataset. The decreasing NLL and increasing accuracy from time 0 (source) to 1 (target) suggest that the probability path from the source to the target distribution gradually shifts the output from the probability space of the source model to that of the ensemble target model.
>
> Table 2. Test losses (NLL & ACC) on the probability path of DBN ($t=0$ to $1$).
>
> |     | DE-1   | t=0.0  | t=0.2  | t=0.4  | t=0.6  | t=0.8  | t=1.0  | DE-3   |
> |-----|--------|--------|--------|--------|--------|--------|--------|--------|
> | NLL | 0.3927 | 0.3927 | 0.3018 | 0.2583 | 0.2417 | 0.2405 | 0.2489 | 0.2221 |
> | ACC |  91.42 |  91.42 |  92.00 |  92.40 |  92.61 |  92.88 |  92.96 |  93.10 |
>
> > I'm not clear how useful the source distribution construction is when its just annealing T. This preserves relative ordering over classes, which is not adding much additional information about the "distribution" of z_1. Are we in some sense integrating over paths from the set of temperature-augmented z_1 -> z? Is that a non-trivial path?
>
> Thanks for the good question. As you said, random temperature annealing does not provide additional information on the ordering over classes, but it offers categorical distributions smoothed at various levels. While there are several methods to provide both additional information and the distribution construction, it has been challenging empirically to find an appropriate approach. For more detailed explanations, please refer to __Common Response 3__.

---

> ### Author Response · Authors · 2023-11-17
> **Response to Reviewer bFwN (2)**
>
> > In algorithm 1: is a single temperature value drawn for each member, for all data points in the minibatch?
>
> While in theory the temperature should be independently drawn for each member, we used a single temperature over each minibatch and yet achieved good performance. We believe that since our diffusion bridge learns over a large number of datasets of hundreds of epochs, the effect of imposing independent temperature over each member or each minibatch does not give rise to a meaningful gap in performance. We will clarify this more clearly in the manuscript.
>
> > Does performance increase with multiple diffusion steps? What's the "cost" of each diffusion step in term of % of DE inference speed. Can DBN admit more than diffusion step at inference and still be faster than DE?
>
> Thank you for your insightful questions.
> In our method, the relative cost (relative FLOPs to the teacher network) of each diffusion step is x0.166 for the evaluation in CIFAR-10/100 datasets, x0.209 for TinyImageNet, and x0.244 for additional ImageNet64 experiments. Applying the diffusion distillation method, only one step of the score network evaluation is required. Hence, the total cost is one forward of the teacher and the one diffusion step of the score, which is x1.166, x1.209, and 1.244, respectively, when we use only a single DBN bridge. If we use more than one step, for example 5 steps, then the total cost is x1.830, x2.045, and x2.220, respectively, which are not competitive anymore.
>
> Table 3. DBN on CIFAR-10 with different number of inference steps. cNLL stands for NLL calibrated with an optimal temperature.
>
> |                  | ACC   | NLL    |
> |------------------|-------|--------|
> | DE-1             | 91.30 | 0.3382 |
> | +1 DBN (step 5)  | 92.97 | 0.2435 |
> | +1 DBN (step 10) | 93.01 | 0.2441 |
> | +1 DBN (step 15) | 92.95 | 0.2442 |
>
> __Table 3__ shows the performance for various diffusion steps. This suggests that the performance of the diffusion bridges converges to the optimal even with a relatively small number of sampling steps. We think, in contrast to diffusion models for high-dimensional image, video, or text datasets, our model, focused on reconstructing ensemble output, aims to identify the optimal path from the source to target logits. The "nominated" logits, potentially close to the target label, are comparatively sparse among all labels, resulting in a sparsified path. Consequently, the role of the diffusion bridge is to find this sparse, low-dimensional path, making the task much simpler than high-dimensional generation tasks.
>
> > DBN models are notably less calibrated than ensembles or bridge networks across tasks. Can the authors comment on the calibration error increase with this method?
>
> According to the results in Table 1 of the paper, the expected calibration error (ECE) score is higher (worse) compared to the BN paper but outperforms existing ensemble distillation methods like ED [1] or END2 [2]. Upon closer examination of these seemingly contradictory results, we arrived at the following interpretations and explanations.
>
> The ECE metric inherently faces limitations in measuring the exact calibration error. In contrast to other calibration metrics like NLL and BS, which directly estimate the level of generalization through continuous calculation, the ECE score initially divides the confidence space into discretely defined bins. The artificial and discrete division of confidences over the dataset introduces some inaccuracy in determining the "exact" calibration level. This is due to the fact that (a) data points in the same bin are treated the same, even if there still exists a calibration error at that specific data point, and (b) the neighboring bin is considered to be disjoint.
> Nonetheless, DBN still takes advantage of the generalization performance with respect to the NLL and Brier score(BS), which also capture the performance of the uncertainty calibration.
>
> [1] J. Hinton et al.,“Distilling the knowledge in a neural network”, NIPS 2015 \
> [2] M. Ryabinin et al., “Scaling ensemble distribution distillation to many classes with proxy targets”, NeurIPS 2021
>
> __Misc typos__
>
> > Section 4.1 "?? of ??" still in text
>
> We sincerely apologize for the error. We fixed this in the revised manuscript.

---

### Official Review · Reviewer_unod · 2023-11-09

**Soundness:** 2 fair
**Presentation:** 4 excellent
**Contribution:** 2 fair
**Rating:** 5
**Confidence:** 4

**Summary:**

The paper proposes to use Diffusion Schrodinger Bridge (DSB) as a way of bridging two arbitrary distributions, namely that of the output activations space of deep models, thereby achieving more efficient ensemble. In contrast to the previous Bridge Network approaches, the DSB approach imposes no (direct) restriction on the low-loss subspace of the parameter manifold, but instead learn a diffusion that solves the SB problem to diffuse from any ensemble member to the "target" ensemble model. The authors claim that the empirical results demonstrate superior performance and FLOPs requirement of the Diffusion Bridge Network (DBN).

**Strengths:**

- The paper generally tackles the important issue of accelerating/simplifying the costly deep ensemble process.
- The overall idea is clearly presented, and it's interesting to motivate a generative solution on the ensemble problem.
- Relatively comprehensive comparison to baseline approaches.

**Weaknesses:**

- The baseline performances shown in the experiment section does not completely agree with the prior works. The scale of the experiments were somewhat small.
- Lack of analysis on the diffusion part of the design.
- The methodology somewhat concerns me (see questions below).

**Questions:**

1. The introduction of temperature does define a distribution, but essentially at a cost of compromising the performance of the original ensemble member (for example, if you look at the NLL loss of the logits after applying temperature, it will be worse). Is this really a reasonable design? If you look for bridging on logit space with another, wouldn't it make more sense to consider the logit space as is (e.g., get a distribution via input perturbation, etc.)?

2. Why DSB rather than a conditional diffusion from Gaussian? In $\textsf{I}^2\textsf{SB}$ the motivation was clearer as they hoped to leverage the structural prior of the image. Empirically, how big would the difference be?

3. The paper has a bunch of analysis on the ensemble approach, but little (almost none) on the diffusion portion. For example, how much training (data and speed) is needed for the DSB training? Why diffusion instead of some simpler approaches like normalizing flows, and bringing all ensemble members to a (same) tractable distribution?


--------------

Post-rebuttal edit: I'm raising my score slightly from 3 to 5, in light of the amendments the authors make during the rebuttal. However, I still believe this paper is not completely ready for publication at the venue. See comment below for details.

4. About the baseline: the Bridge Network paper seems to have better numbers on TinyImageNet than what the authors reported in this paper (e.g., 65.82% accuracy for 3-BN-sm and the FLOPs seems to be much smaller (1.15x). The BN paper also evaluated on ImageNet, where in my experience the accuracy is **much less volatile**. If, as authors claim that the BN approach has more training cost (at the end of Sec. 2.1), perhaps the DSB approach should be compared with BN on that turf as well.

---

> ### Author Response · Authors · 2023-11-17
> **Response to Reviewer unod (1)**
>
> We appreciate your sincere consideration and insights on our paper! We address the raised issues in detail as outlined below.
>
> __On the rightness of reproducing the baseline model and extended experiments__
>
> > Weakness\
> The baseline performances shown in the experiment section does not completely agree with the prior works. The scale of the experiments were somewhat small.
>
> > Questions\
> About the baseline: the Bridge Network paper seems to have better numbers on TinyImageNet than what the authors reported in this paper (e.g., 65.82% accuracy for 3-BN-sm and the FLOPs seems to be much smaller (1.15x). The BN paper also evaluated on ImageNet, where in my experience the accuracy is much less volatile. If, as authors claim that the BN approach has more training cost (at the end of Sec. 2.1), perhaps the DSB approach should be compared with BN on that turf as well.
>
> Thank you for the careful investigation of our experiments. We double-checked our experiments and figured out that the saved checkpoint of the teacher network for training BN had a problem. Precisely, one of the layers of the teacher network of BN was saved in the different layer and it caused the performance drop. We fixed this and trained the BN again. And as you pointed out that the performance is much less volatile for large-scaled datasets, we tried our method in the ImageNet dataset. Because of the limit of the training budget, we tried this on the downscaled $64\times 64$ datasets.
> Since this issue is the core part of our response that can strengthen our method, we organized our experimental results for extended datasets in the __Common Response 1__.
>
> * * *
>
> __Analysis on the design of the diffusion part__
>
> > Questions\
> Why DSB rather than a conditional diffusion from Gaussian? In I2SB  the motivation was clearer as they hoped to leverage the structural prior of the image. Empirically, how big would the difference be?
>
> Thanks for your insightful question! Just like I2SB aims to leverage the structural prior of the image from *degraded* images, our paper shares a similar motivation. We aim to exploit the *degraded* logits generated from a single classifier as the structural prior of the logit space of the ensemble. This implies that the functional outputs of the ensemble members can be represented as a distribution conditioned on those of the (single) source member.
>
> The conditional diffusion from Gaussian can also yield the output logits, but this does not utilize the useful information of the logit of the source model. This is also shown in the image case: I2SB, which constructs the diffusion bridge from the degraded source image distribution to the fine-grained target distribution, outperforms the previous conditional diffusion model approach [2] which generates the fine-grained images by imposing the degraded source images as its condition. We can achieve good performance with a small number of diffusion steps because we can leverage information from the source model. If the same performance is desired in the conditional diffusion, it would require a greater number of diffusion steps. To support this claim, we trained the conditional diffusion with the same number of diffusion steps as I2SB and Table 1 shows an experiment that compares I2SB with the conditional diffusion method.
>
> Table 1. I2SB vs. conditional diffusion from Guassian on CIFAR10.
>
> |                         | ACC   | NLL    | DEE   |
> |-------------------------|-------|--------|-------|
> | DE-1                    | 91.30 | 0.3382 | 1.000 |
> | +1 DBN (I2SB)           | 92.97 | 0.2435 | 2.228 |
> | 1 conditional diffusion | 90.78 | 0.3727 |   < 1 |
>
> [1] G. Liu et al., “I2SB: Image-to-Image Schrödinger Bridge”, ICML 2023, URL: https://arxiv.org/abs/2302.05872 \
> [2] C. Saharia et al., “Palette: Image-to-Image Diffusion Models”, SIGGRAPH 2022, URL: https://arxiv.org/abs/2111.05826

---

> ### Author Response · Authors · 2023-11-17
> **Response to Reviewer unod (2)**
>
> > Questions \
> The paper has a bunch of analysis on the ensemble approach, but little (almost none) on the diffusion portion. For example, how much training (data and speed) is needed for the DSB training? Why diffusion instead of some simpler approaches like normalizing flows, and bringing all ensemble members to a (same) tractable distribution?
>
> Thank you for your thoughtful question of pointing out the diffusion part of our method. The precise required training resource will be added in the appendix, in place of the architecture, and hyperparameter setting section.
>
> Recognizing the performance limitations of BN, we felt the need for a more flexible approach. Therefore, instead of simple methods like the conventional discrete-time Normalizing Flow (NF), which has limitations in distribution approximation, requiring invertible networks for likelihood maximization, we adopted diffusion-based methods such as I2SB. In the case of methods like NF, creating a tractable distribution necessitates the use of invertible networks, imposing constraints on the network structure. In contrast, diffusion-based methods do not have such restrictions on network invertibility, allowing for the generation of a more flexible probability path. Despite the relative complexity of I2SB in the theoretical background, it delivered sufficient performance with low computational cost. Of course, as an alternative, one could also choose a slightly more flexible method yet similar to NF, such as Rectified Flow (RF). Its experimental results are described in the __Common Response 2__.
>
> * * *
>
> __Methodology__
>
> > Weakness \
> The methodology somewhat concerns me (see questions below).
>
> > Questions \
> The introduction of temperature does define a distribution, but essentially at a cost of compromising the performance of the original ensemble member (for example, if you look at the NLL loss of the logits after applying temperature, it will be worse). Is this really a reasonable design? If you look for bridging on logit space with another, wouldn't it make more sense to consider the logit space as is (e.g., get a distribution via input perturbation, etc.)?
>
> Thank you for the thoughtful responses and feedback! According to the question, we provided a precise explanation on why we have introduced the temperature annealing in __Common response 3__. Moreover, we added missing details on how the random temperature annealing is defined. For our choice of the temperature distribution, we chose the Beta distribution $T = 2 ( 1+0.2\alpha), \alpha\sim \mathrm{Beta}(\cdot; 1, 5)$ for simple and efficient temperature annealing. We added these details on hyperparameters including the temperature annealing in Appendix B.

---

> ### Comment · Reviewer_unod · 2023-11-20
> **Response to the authors' rebuttal**
>
> I'd like to thank the authors for the detailed response, explanations and clarifications in the rebuttals. It is nice that the authors have been able to provide some (preliminary) results on ImageNet 64x64 and correct the DBN vs. BN quantitative difference issue via model debugging.
>
> However, after reading the reviews of other fellow reviewers and the rebuttals, some of my concerns remains. Specifically:
>
> 1. I have consistently found some arguments to be still hand-wavy and lack sufficient either empirical analysis or ablative analysis.
>
>     a) For example, on the temperature issue (which most reviewers have raised), I am still not convinced that "random temperature annealing is an easy and simple way to construct the source distribution". Unlike the in multinomial distribution sampling case (e.g., language modeling or VQ) where temperature encourages sampling diversity, I am still unsure why the temperature-annealed distribution is still the same as output distribution of the logits. E.g., if your temperature is really high, then the distribution will approach uniform--- which probably won't be the model output distribution at all. I think this will worth a fair amount of ablative analysis to motivate the design. Moreover, while I appreciate the observation notes that the authors make on the mixup experiments, I am surprised if this is a "cliff function" behavior (i.e., weak mixup = no distribution shift; heavy mixup = hard to learn) and that there is no middle ground (e.g., on the mixup ratio spectrum). What is the behavior like? I think these questions are much better answered with actual plots, ablation experiments, visualizations, gradient analysis, etc. If the authors were to really propose DBN as a novel approach of the ensembling, I think it will be greatly consolidated by all these accompanying experiments, in addition to the "best accuracies".
>
>     b) As another example, I think the result that the authors obtained on the flow experiment exactly corroborated what I feared. Specifically, on CIFAR-10, the accuracy of 92.89 and 92.97 is well within model variation (e.g., from SGD or param init). Rather than claiming one is better, I'd say they are more "on par". I think we need more evidence when the authors claim that "We suggest that this performance gain is due to the robustness obtained by the stochasticity...". Moreover, this new experiment probably implies that we probably don't need a distribution mapping approach as sophisticated as DSB, but a simpler change-of-variable based approach like flow-based method.
>
> 2. I acknowledge that reproducing another paper could be challenging sometimes (e.g., currently, it seems that the result of DBN is still worse than the BN small (65.82) and medium (66.76) in the original Yun et al. paper). However, the current Table 1 (amended by the rebuttal Table 1) kind of suggests that the BN method is worse on both accuracy **and** FLOPs, which is not entirely correct, because there's also BN_{small} that was 1.15x FLOP and (at least in the original paper) in the 64.5-65.8 range for accuracy. The authors should fairly list them in your Table 1 as well.
>
> I'm raising my rating to reflect the positive changes and experiments that the authors did in the rebuttal (which I deep appreciate). However, due to the aforementioned issues, I still feel this paper lacks sufficient analysis, experiments, and comparisons to bolster many of the key claims.

---

> ### Author Response · Authors · 2023-11-21
> **Re: Response to Reviewer unod (1)**
>
> __1 (a) On the temperature issue__
>
> Thanks for the good question. As you mentioned, when $T$ approaches to infinity, the categorical source distribution becomes uniform. Just as Knowledge Distillation requires appropriate temperature choice, preventing it from focusing on label class only ($T=1$) or erasing all information ($T=\infty$), an appropriate temperature distribution, maybe ranging from 1 to 20, is necessary. We obtained good results with $T = 2(1+0.2\gamma)$, where $\gamma$ is sampled from a Beta distribution with parameters $1$ and $5$.
>
> To verify that the random temperature annealing is a valid choice, We ablated our proposed temperature annealing method to the mixup approach, as we have explained in the rebuttal. With varying mixup parameters $\alpha$ in mixup formula, $\boldsymbol{x}' = \lambda \boldsymbol{x}_1 + (1-\lambda)\boldsymbol{x}_2$, where $\lambda$ is sampled from a Beta distribution with parameters $\alpha$ and $\alpha$, with values of 0.1, 0.3, 0.5, 0.7, and 0.9, the results are demonstrated in __Table 2__ as follows:
>
> Table 2. DBN in CIFAR-10 with different mixup $\alpha$ on the source distribution.
> |                        | ACC   | NLL    |
> |------------------------|-------|--------|
> | DE-1                   | 91.30 | 0.3382 |
> | +1 DBN (alpha 0.1)     | 92.47 | 0.2632 |
> | +1 DBN (alpha 0.3)     | 92.53 | 0.2596 |
> | +1 DBN (alpha 0.5)     | 92.75 | 0.2579 |
> | +1 DBN (alpha 0.7)     | 92.62 | 0.2613 |
> | +1 DBN (alpha 0.9)     | 92.29 | 0.2715 |
> | +1 DBN (rand. anneal.) | 93.04 | 0.2394 |
>
> the following aspects are observed:
> * Without using the random temperature annealing, $\alpha=0.5$ achieved the best performance among all possible choices. It shows a significant drop around $\alpha=0.9$. *(Cliff behavior)*
> * Even the best working mixup did not reach the random temperature annealing as shown in __Table 2__. *(Advantage of random annealing)*

---

> ### Author Response · Authors · 2023-11-21
> **Re: Response to Reviewer unod (2)**
>
> __1 (b) Comparison to the generative flow approach__
>
> Thank you for the thoughtful feedback on our ablation study. In fact, there exists a bunch of literature suggesting the diffusion Schrodinger bridge serves as a generalization of the generative flow methods. These methods simulate through an ODE constructed to match two different distributions, i.e., the (continuous) normalizing flow or the rectified flow can be interpreted as a special case of Schrodinger bridge. For instance, [1] (Section 3.3) have investigated that the diffusion Schrodinger bridge (DSB) converges to the optimal transport plan with respect to the 2-Wasserstein distance [2, 3] as the diffusion coefficient $\beta_t$ approaches zero. Since DSB is exploited regardless of the value of $\beta_t$ (if it is zero, then DSB becomes a flow), the flow-based model can be viewed as a special case of DSB, not a specifically different one.
>
> From a different perspective in terms of the partial differential equations (PDE) that is shared by both SDE and ODE (Fokker-Planck equation), [4] proposes a framework that unifies flow-based and diffusion-based methods. According to this unified perspective, a single step of the practical sampler of the stochastic sampling (via SDE) is just an addition to the deterministic sampler of the deterministic sampling (via ODE) added by some Gaussian noises, such as
> $$
> \frac{d}{dt} Z_t = b(t, Z_t) \\
> d Z_t = b(t, Z_t) dt + \beta_t \eta (t, Z_t) dt + \sqrt{2 \beta_t} d W_t
> $$
> where $b(t, Z_t)$ and $\varepsilon$ corresponds to the diffusion coefficient and $\eta$ corresponds to the additional denoiser term added in the stochastic sampler, that can be simply obtained by solving the Fokker-Planck equation. (For further details, please refer to Figure 3 of page 10 of [4]) The deterministic sampling via ODE corresponds to the (continuous) normalizing flow [5] (or rectified flow [6], the free-form (e.g. does not require deliberately designed invertible architecture) likelihood-based generative model.
>
> The conventional normalizing flows (e.g. Real NVP, Glow) can also be regarded but it has intrinsic limitation (a single invertible network) compared to the continuous normalizing flows in terms of approximation of two distributions so we **excluded from the candidates**.
>
> Subsequently,
> - **The flow-based model is a special case of diffusion bridge; using I2SB or continuous normalizing flow can be regarded as a hyperparameter tuning of $\beta_t$ which solves the same distribution matching problem.**
> - **In the optimal transport perspective, the Schrodinger bridge models converge to the flow-based models, when the noise coefficient $\beta_t$ converges to zero.**
> - **In the partial differential equation perspective, flow and diffusion-based models result in the equivalent solutions, and their solvers have the same form except for some Gaussian noise.**
>
> [1] G. Liu et al., “I2SB: Image-to-Image Schrödinger Bridge”, ICML 2023 \
> [2] T. Mikami et al., “Monge’s problem with a quadratic cost by the zero-noise limit of h-path processes”, Probability theory and related fields 2004 \
> [3] G. Peyre et al., “Computational Optimal Transport”, Foundations and Trends in Machine Learning 2019 \
> [4] M. Albergo et al., “Stochastic interpolants: A Unifying Framework for Flows and Diffusions”, arXiv:2303.08797 \
> [5] W. Grathwohl et al., “FFJORD: Free-Form Continuous Dynamics for Scalable Reversible Generative Models”, ICLR 2019 \
> [6] X. Liu et al., “Flow Straight and Fast: Learning to Generate and Transfer Data with Rectified Flow”, ICLR 2023
>
> * * *
>
> __2. Reproducilibity of the BN__
>
> If you check the source code for the BN [7], you'll find that it uses the indices from 0 to 90,000 from the TinyImageNet training set as the training data and uses indices from 90,000 to 100,000 as the validation data. In contrast, in our work [8], we used indices from 0 to 81,920 as the training data and indices from 81,920 to 100,000 as the validation data. Due to these differences, despite synchronizing the other hyperparameters with the BN source code, we think that it has led to an overall performance degradation in our work. Therefore, as mentioned, we are currently conducting experiments with BN$_\mathrm{small}$, and we will present the results as soon as they are available.
>
> [7] https://github.com/yuneg11/Bridge-Network/blob/main/configs/tin200/modules/dataset.yaml \
> [8] Supplementary Material, “data/build.py” line 198.

---

> ### Author Response · Authors · 2023-11-21
> **Re: Response to Reviewer unod (3)**
>
> Here is the results including BN$_\mathrm{small}$:
>
> Table 3. Test results on TinyImageNet.
> |          | FLOPs  | #Params | ACC   | NLL    | cNLL   | DEE   |
> |----------|--------|---------|-------|--------|--------|-------|
> | DE-1     | x1.000 | x1.000  | 59.26 | 1.8399 | 1.8041 | 1.000 |
> | +2 BN_sm | x1.099 | x1.114  | 63.08 | 1.5572 | 1.5531 | 2.854 |
> | +3 BN_sm | x1.149 | x1.171  | 63.44 | 1.5258 | 1.5171 | 3.556 |
> | +2 BN_md | x1.359 | x1.412  | 64.07 | 1.5066 | 1.4938 | 4.102 |
> | +1 DBN   | x1.196 | x1.149  | 64.60 | 1.5299 | 1.5296 | 3.444 |
> | +2 DBN   | x1.391 | x1.298  | 65.34 | 1.4890 | 1.4865 | 4.701 |
> | DE-2     | x2.000 | x2.000  | 62.49 | 1.6219 | 1.6257 | 2.000 |
>
> As you mentioned, “+3 BN_sm” has slightly better DEE (*3.556*) compared to that of  “+1 DBN” (*3.444*). However, we want to emphasize that the accuracy of “+1 DBN” (**64.60**) significantly outperforms that of “+3 BN_sm” (**63.44**). Moreover, due to the fast performance saturation of BN (also shown in Figure 4 of the paper), DBN exhibits better performance than BN in both ACC and NLL as the cost budget (FLOPs,#Params) increases.
>
> Despite the inferior measurement of DE-1 performance in our setting, the relative performance (DEE) of BN_sm and BN_md appears quite similar. This suggests that BN is adequately reproduced in our setting, and we consider the comparison between DBN and BN in the current setting to be fair.

---

### Author Response · Authors · 2023-11-17
**Overall response to all reviewers (2)**

__Common response 3 - The temperature scaling in the source data issues__

We empirically discovered that without introducing a random temperature to add stochasticity to the source distribution, the path from the source logit space to the target tends to lead to some trivial solutions through the diffusion bridge. Even though the source and target distributions come from different functional outputs of ensemble members, they do not vary significantly. Since our task involves extracting information from diverse ensemble members at once, we added stochasticity to the source domain, which prevents it from falling into trivial solutions, and also does not harm the source information.

As the reviewer unod pointed out, there is a possibility of achieving robustness by introducing input perturbations to get the distributions. Indeed, we experimented with a perturbation through the mixup augmentation and found that the following phenomena prevented the input perturbation from obtaining the well-designed robust distributions.

1. When we applied a *weak* input perturbation, it did not result in a meaningful distribution shift in the logit space and did not enhance robustness compared to our proposed random temperature annealing approach.

2. When we applied a *heavier* input perturbation, it increased the discrepancy between source and target distribution, making the learning of the diffusion bridge more difficult and causing the learned diffusion bridge to perform worse than the proposed approach.

One of the important things is that the random temperature annealing is an easy and simple way to construct the source distribution compared to the other methods including input perturbation, random model selection, and so on. For these reasons, we chose the random temperature annealing to give robustness in the source domain. We will clarify the method section by reflecting those explanations.

* * *

__Common response 4 - Clarification of writing (especially for the diffusion Schrödinger bridge, and the details on our method)__

We sincerely apologize for confused writing and grammar errors. We revised our manuscript by clarifying the points that the reviewers have issued. The following aspects of our papers were refined.

* We complemented Section 2.2 (the background on diffusion Schrödinger bridge) by including the equation of the coupled PDE for the solution pair of the Schrödinger equation and added the denotations of the variables and coefficients we have initiated in the paper. Please refer to Section 2.2 (colored blue) in the paper.

* We fixed some denotations in the main method and briefly added how Equation (9) (the reverse SDE used for the sampling process) is derived. Please refer to Section 3.2 (colored blue) in the paper.

* We added the inference algorithm of our method in the Appendix, which was omitted in the main paper. Please refer to Section C of our appendix.

* We supplemented the detailed architectures, training details, and hyperparameter settings in the Appendix. Please refer to Section B in our appendix.

---

### Author Response · Authors · 2023-11-17
**Overall response to all reviewers (1)**

We thank all reviewers for their valuable comments. We are excited that the reviewers identified the novelty of our contribution, appreciated our experimental results and acknowledged the significance of our work. We also appreciate all reviewers for thoughtful reading and feedback which further improve on our work.

__Common response 1 - Extended experiments (Reviewer unod, bFwN)__

> Corrected results on TinyImageNet.

We sincerely appreciate Reviewer unod for the careful investigation of our experiments. We double-checked our experiments and figured out that the saved checkpoint of the teacher network for training BN had a problem. Precisely, one of the layers of the teacher network of BN was saved in the different layer and it caused the performance drop. We fixed this and trained the BN again, and obtained the following result. To be precise, the following are different from the experimental results in the main paper.
* The performance of +2 BN is corrected, with inference over the correct checkpoint.
* The performance of +2 DBN is fine-tuned with further hyperparameter-searching.
* The ratio of #Params w.r.t to DE-1 baseline of +2 DBN was reported as 1.398, but this is corrected to __1.298__. This is natural since the FLOPs and number of parameters added from DE-1 should be doubled from +1 DBN.
We will reflect the revision in the final version.

Table 1. Corrected results on TinyImageNet. cNLL stands for NLL calibrated with an optimal temperature. (We also fixed the typo in the Table 1 in the main paper; #Params of “+2 DBN”: $\times1.398 \to \times1.298$)
|        | FLOPs  | #Params | ACC   | NLL    | cNLL   | DEE   |
|--------|--------|---------|-------|--------|--------|-------|
| DE-1   | x1.000 | x1.000  | 59.26 | 1.8399 | 1.8041 | 1.000 |
| +2 BN  | x1.359 | x1.412  | 64.07 | 1.5066 | 1.4938 | 4.102 |
| +1 DBN | x1.209 | x1.149  | 64.60 | 1.5299 | 1.5296 | 3.444 |
| +2 DBN | x1.418 | x1.298  | 65.34 | 1.4890 | 1.4865 | 4.701 |
| DE-2   | x2.000 | x2.000  | 62.49 | 1.6219 | 1.6257 | 2.000 |

> Results on downscaled ImageNet.

As the reviewers were concerned about the scalability of our method, we tried to extend our method in the ImageNet dataset. Because of the limit of the training budget, we conducted this on the downscaled $64\times 64$ datasets and obtained the following result. We also attached the results of BN in the baseline ImageNet and compared the DBN and BN with the baseline DEs independently. Then we observed that the performance of our DBN method surpasses DE-2 in terms of DEE and accuracy (ACC) and it is close to that of DE-3 with only one diffusion bridge, while BN achieves the same level of performance with two bridges. We will add full results of ImageNet in the final version.

Table 2. Results on ImageNet(64x64).

|        | FLOPs  | #Params | ACC   | cNLL   | DEE   |
|--------|--------|---------|-------|--------|-------|
| DE-1   | x1.000 | x1.000  | 66.47 | 1.3960 | 1.000 |
| +1 DBN | x1.244 | x1.380  | 69.79 | 1.2140 | 2.702 |
| DE-2   | x2.000 | x2.000  | 69.56 | 1.2542 | 2.000 |
| DE-3   | x3.000 | x3.000  | 70.82 | 1.1969 | 3.000 |

Table 3. Results on ImageNet(224x224), reported in the BN paper.

|       | FLOPs  | #Params | ACC   | cNLL  | DEE   |
|-------|--------|---------|-------|-------|-------|
| DE-1  | x1.000 | x1.000  | 75.85 | 0.936 | 1.000 |
| +1 BN | x1.194 | x1.222  | 77.03 | 0.889 | 1.881 |
| +2 BN | x1.389 | x1.444  | 77.37 | 0.876 | 2.341 |
| DE-2  | x2.000 | x2.000  | 77.12 | 0.883 | 2.000 |
| DE-3  | x3.000 | x3.000  | 77.64 | 0.862 | 3.000 |

* * *

__Common response 2 - Compare to the generative flow approach (Reviewer unod, jMzV)__

Our approach constructs a probability flow by generating the diffusion bridge between two distributions, a source distribution of the single network output, and a target distribution that consists of ensemble outputs. Generative flows such as normalizing flow, rectified flow, or flow matching can be good alternatives to our diffusion bridge, as the reviewer jMzV mentioned the benefits of variance reduction in deterministic flow approaches. To verify the consistency of our approach in the generalized flows, we extended our experiments to Rectified flow (RF) in CIFAR-10 in __Table 4__. We observed that DBN with I2SB slightly outperforms DBN with RF. We suggest that this performance gain is due to the robustness obtained by the stochasticity of the flow, compared to the RF method. However, we note that RF or other generative flow approaches can be utilized as good alternatives to I2SB, as these deterministic methods require less subtle hyperparameter optimization at a cost of the flexibility.

Table 4. I2SB vs. Rectified Flow (RF) on CIFAR10.

|               | ACC   | NLL    | DEE   |
|---------------|-------|--------|-------|
| DE-1          | 91.30 | 0.3382 | 1.000 |
| +1 DBN (I2SB) | 92.97 | 0.2435 | 2.228 |
| +1 DBN (RF)   | 92.89 | 0.2448 | 2.173 |

---

### Meta-Review · Area_Chair_ECpN · 2023-12-06

**Metareview:**

This paper proposes a computationally cheaper alternative to deep ensembles, where the ensemble predictive distribution is approximated using a Schrödinger bridge from a single network predictive. After an active discussion between authors and reviewers, the overall assessment is rather positive, with four reviewers leaning towards acceptance (two of them strongly so), while only one (weakly) leans towards rejection. While the reviewers praised the importance of the problem, the comprehensive experiments, and strong empirical performance, they were critical of the lack of ablation studies on the diffusion approach, the unclear motivation for the temperature annealing, the uncertainty calibration, and the clarity of writing and experimental details. However, most (if not all) of these issues have been addressed in the extensive rebuttal. It therefore seems warranted to accept the paper, with the understanding that the authors will continue their efforts to fully address the reviewer feedback in the camera-ready version.

**Justification For Why Not Higher Score:**

It is unclear whether this paper is interesting to a wide enough audience to warrant a spotlight

**Justification For Why Not Lower Score:**

The authors were very active during the rebuttal period and managed to convince most of the reviewers of the merits of the paper

---

### Decision · Program_Chairs · 2024-01-16

Accept (poster)